# *In silico* generation of synthetic cancer genomes using generative AI

## Graphical abstract

## Authors

Ander Díaz-Navarro, Xindi Zhang, Wei Jiao, Bo Wang, Lincoln Stein

## Correspondence

ander.d.navarro@gmail.com (A.D.-N.),
lincoln.stein@gmail.com (L.S.)

## In brief

Díaz-Navarro et al. present OncoGAN, a tool that generates realistic synthetic cancer genomes simulating tumor-specific features and patterns. These genomes contain known mutations and alterations, providing a valuable resource for benchmarking and improving cancer genome analysis tools. Moreover, OncoGAN preserves real donors' privacy, making the simulations fully accessible.

## Highlights

- OncoGAN is a multimodel ensemble pipeline designed to generate synthetic cancer genomes

- Key features like mutational signatures, genomic patterns, or CNA-SV are modeled

- Features are simulated to match eight tumor-specific profiles

- Synthetic donors have no access restrictions and preserve real donors' privacy

Díaz-Navarro et al., 2025, Cell Genomics 5, 100969
November 12, 2025 © 2025 The Authors. Published by Elsevier Inc.

CellPress

## Article

# *In silico* generation of synthetic cancer genomes using generative AI

Ander Díaz-Navarro,[1,2,*] Xindi Zhang,[1] Wei Jiao,[1] Bo Wang,[3,4,5,6,7] and Lincoln Stein[1,2,5,8,*]
[1]Ontario Institute for Cancer Research, Toronto, ON, Canada
[2]Department of Molecular Genetics, University of Toronto, Toronto, ON, Canada
[3]Peter Munk Cardiac Centre, University Health Network, Toronto, ON, Canada
[4]Department of Computer Science, University of Toronto, Toronto, ON, Canada
[5]Vector Institute, Toronto, ON, Canada
[6]Department of Medical Biophysics, University of Toronto, Toronto, ON, Canada
[7]Department of Laboratory Medicine and Pathobiology, University of Toronto, Toronto, ON, Canada
[8]Lead contact
*Correspondence: ander.d.navarro@gmail.com (A.D.-N.), lincoln.stein@gmail.com (L.S.)

## SUMMARY

Understanding how genomic alterations drive cancer is key to advancing precision oncology. To detect these alterations, accurate algorithms are used; however, due to privacy concerns, few deeply sequenced cancer genomes can be shared, limiting benchmarking and representing a major obstacle to the improvement of analytic tools. To address this, we developed OncoGAN, a generative AI model combining adversarial networks and variational autoencoders to create realistic synthetic cancer genomes. Trained on large-scale genomic datasets, OncoGAN accurately reproduces somatic mutations, copy number alterations, and structural variants across cancer types while preserving donors' privacy. The synthetic genomes reflect tumor-specific mutational signatures and positional mutation patterns. Using DeepTumour, we validated the synthetic data's fidelity, showing high concordance between generated and predicted tumors. Moreover, augmenting the training data with synthetic genomes improved DeepTumour's accuracy, underscoring OncoGAN's potential to generate shareable datasets with known ground truths for benchmarking and enhancement of cancer genome analysis tools.

## INTRODUCTION

The vast majority of cancers arise from genomic damage that alters the activity of key genes and proteins involved in the pathways that regulate cell proliferation, programmed cell death, and interaction with surrounding tissues.[1] The promise of precision oncology is that by cataloging these genomic alterations, clinicians can apply therapies that target these altered pathways for enhanced patient outcomes and reduced side effects. Cancer genome alterations have been extensively explored through multiple large-scale genomic projects, starting with the International Cancer Genome Consortium[2] and The Cancer Genome Atlas[3] and continuing to this day with such large-scale projects as the PanCancer Analysis of Whole Genomes (PCAWG),[4] the 100,000 Genomes Project,[5] or the Hartwig Medical Foundation.[6] Collectively, these projects have analyzed over 20,000 whole genomes from patients with different types of tumors, detecting a large number of point mutations as well as structural variants (SVs) and copy number alterations (CNAs).[5,7–9] Using these extensive datasets, researchers have utilized the frequency of these genomic alterations alongside clinical characteristics to identify the key drivers of the cancer phenotype.

To analyze these data, the research community has developed a series of complex computational pipelines, employing multiple variant "callers,"[10–14] each optimized to better detect a specific range of mutations.[4] However, a fundamental issue with these tools is the absence of a gold standard set of cancer genomes with known mutations to test the callers against, and without such a standard it is difficult to know the true error profile of a pipeline or to rank two pipelines against each other. Computational biologists instead benchmark their mutation calling tools against a small set of approximately 50 legacy genomes derived from cancer cell lines[15,16] or datasets compiled by the Genome in a Bottle Consortium with high-confidence variant calls[17] or, alternatively, they test their tools on normal genomes that have been spiked with a known series of *in silico*-generated mutations.[18,19] Unfortunately, none of these approaches fully captures the mutational diversity of actual patient-derived cancer genomes, which vary widely according to the tissue of origin, treatment history, and environmental exposures. Compounding this issue are the legal and ethical restrictions on sharing human genomic data, which make it logistically difficult to assemble and share large numbers of cancer genomes.[20–22] Together, these challenges have slowed the development of cancer genome analysis tools.

Recent advances in deep learning techniques, such as generative adversarial networks[23] (GANs) and variational autoencoders[24] (VAEs), represent an opportunity to create open



synthetic datasets that can accelerate the improvement of tumor genome analysis tools. Notably, Yelmen et al. demonstrated the utility of GANs in genomics by constructing a model capable of generating high-quality artificial genomes that simulate haplotype segments across different ethnicities.[25] Briefly, GANs consist of two convolutional neural networks working together. The first one, the generator, creates a result from random data, while the second one, the discriminator or critic, attempts to differentiate real data from synthetic data. Through iterative training of both networks, GANs can achieve highly realistic simulations. VAEs, meanwhile, encode input data into a compressed probabilistic latent space before decoding it back. Unlike traditional autoencoders, VAEs introduce randomness by sampling from the latent space, allowing them to generate new, similar data during the decoding process. Although originally developed for image generation, GANs and VAEs have been adapted for tabular data generation through models such as TGAN,[26] CTGAN,[27] CopulaGAN,[27] TableGAN,[28] or CTAB-GAN+[29] for GAN architectures, as well as tabular VAEs (TVAEs)[27] for VAE-based approaches.

Here, we present OncoGAN, a pipeline that uses GANs, TVAEs, and a random sampling approach trained on large datasets of cancer genome sequencing data to generate unlimited, realistic simulated cancer genomes with known ground truths, including point mutations, CNAs, and SV profiles. The fidelity of these simulated genomes was validated using DeepTumour, a tool that predicts the tumor type of origin based on mutational patterns. Additionally, we demonstrated the utility of these simulations to improve DeepTumour's accuracy through training on a combined dataset of real and synthetic donors. Importantly, our approach ensures donor privacy, making the simulations fully accessible. Using OncoGAN, we have generated and released 800 simulated genomes across eight cancer types, which are free of distribution restrictions and ready for use in developing and benchmarking new cancer genome analysis tools.

## RESULTS

### Multimodel ensemble pipeline for synthetic tumor generation

OncoGAN is a comprehensive pipeline designed to generate synthetic tumor samples for eight distinct cancer types (Figure 1; Figure S1). Cancer genomes are characterized by the accumulation of genetic alterations, which can be broadly categorized into "driver" and "passenger" events. Driver mutations directly contribute to cancer development by conferring growth advantages to cells, while passenger mutations are background mutations that are not directly selected. Any individual cancer will have a small number of drivers against a large background of passengers, and a key aspect of cancer genome analysis is to distinguish among these two classes. These mutations can take various forms, including single nucleotide changes, insertions, deletions, and structural rearrangements, all of which contribute to the complexity of cancer genomes. To accurately simulate the complex mutational landscape of point mutations, we trained five different models using the PCAWG,[4] a set of 2,658 whole cancer genomes that have been sequenced and analyzed according to a uniform standard dataset (see STAR Methods).

The models were trained on cancer genome features organized into two primary categories: donor and mutation characteristics. Donor characteristics include: (1) the number of different mutation types (single/di-/trinucleotide polymorphisms [SNPs/DNPs/TNPs], small insertions [INSs], or small deletions [DELs]) and (2) the number and type of driver mutations, also referred to here as drivers' intercorrelation (i.e., driver co-occurrence and mutual exclusion), present in each donor. By training these models independently, we are able to replicate the tumor heterogeneity observed in real data (see the next section). Mutation characteristics cover (3) mutational signature contexts, (4) genomic positions, and (5) variant allele frequencies (VAFs). Since these features are independent and present unique challenges during training, we developed a separate model for each, and these were then integrated together.

We utilized three distinct strategies for training the models: GANs, TVAEs, and random sampling. Due to its high accuracy in capturing and simulating intrafeature relationships, we employed CTAB-GAN+ for six of the nine models: the number of each type of mutation, the drivers for each donor, mutational signature contexts, and CNA-SV profiles. To predict genomic positions, we combined CTGAN and TVAE architectures with a discretization process (see STAR Methods), enabling us to accurately capture mutation density along the genome. Allele frequencies were simulated by first randomly selecting the mean VAF per donor, based on distributions observed in real patients, and then randomly choosing VAFs according to the expected distribution for that particular mean frequency.

These models were then integrated into an automated pipeline, OncoGAN, which combines their results to generate realistic tumor genomes (Figure S1). The pipeline begins by generating the donor characteristics for the desired cancer type. This includes determining the number of each mutation type within each sample (CTAB-GAN+ model 1), the mean VAF of the donor (sampling model), and the CNA-SV profile, which is based on the total number of mutations previously assigned to each donor (CTAB-GAN+ model 2). Subsequently, the donor's sex is assigned according to the incidence sex ratio for the selected cancer type, and the number of mutations for each driver gene is simulated while preserving drivers' intercorrelation (CTAB-GAN + model 3). Since driver mutations cannot be randomly generated, they are selected from a tumor-type-specific list derived from real patient data published previously by the PCAWG.[4,30,31] Following this, the list of passenger mutations and their type (CTAB-GAN+ model 4) and their genomic positions (TVAE model 1) are generated. Given that the models lack reference genome information (i.e., the ability to map coordinates to their DNA sequences), they cannot directly match a generated mutation's trinucleotide reference context to its position. Consequently, after all mutations are simulated, the trinucleotide contexts and mutation positions are aligned by searching for the reference context within the sequence extracted from the coordinates. A window size of 100 nt around the predicted position is used to search for the expected trinucleotide context, and if no match is found, a new position is simulated to ensure minimal deviation from the initial prediction. This strategy protects donor privacy and ensures that any simulated passenger mutations appearing in the training set do so purely by chance. On average, only 0.021%

## Article

CellPress

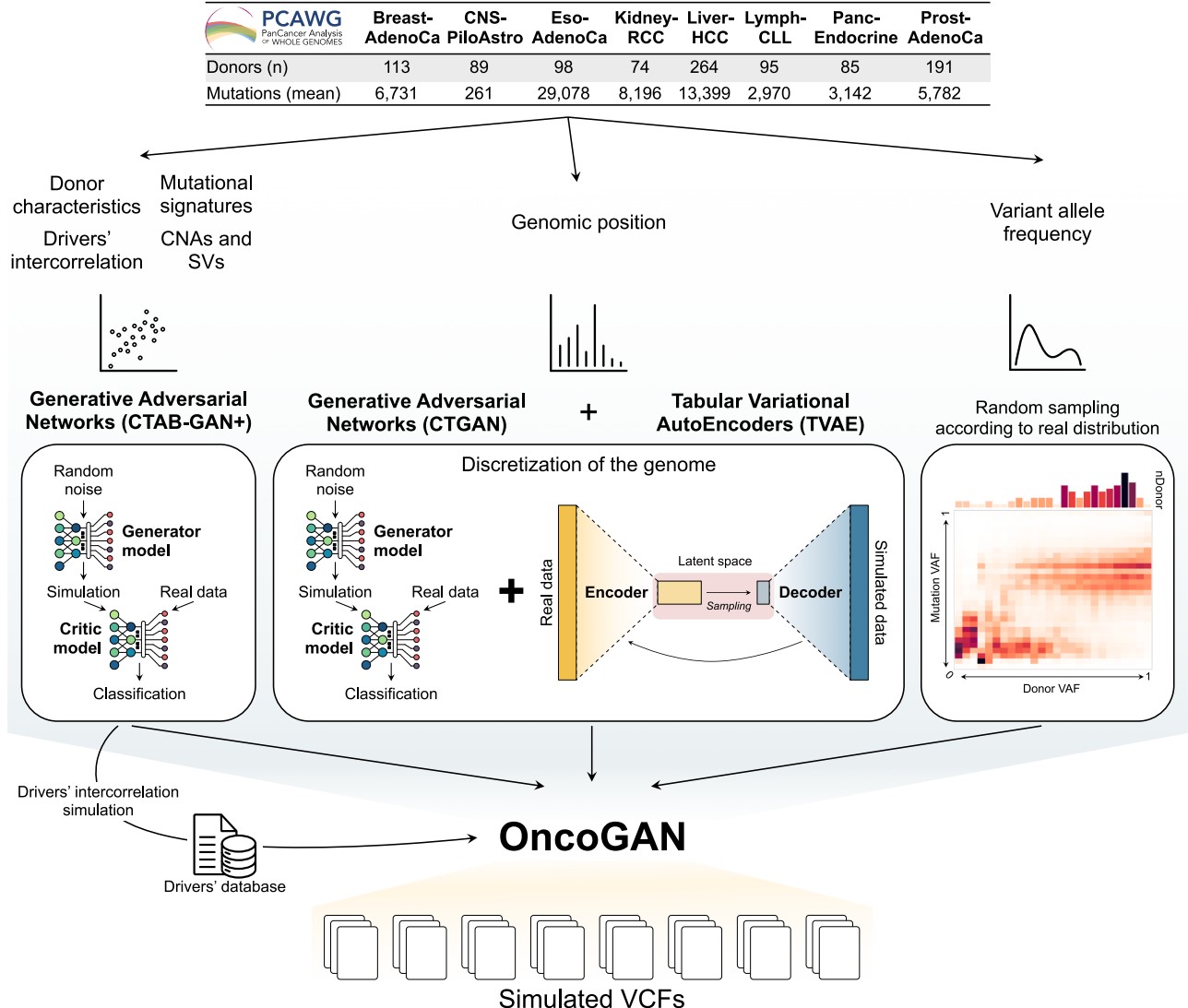

**Figure 1. Overview of the OncoGAN ensemble pipeline for the simulation of synthetic VCFs**

OncoGAN integrates eight distinct models, each trained on data from eight different tumor types sourced from the PCAWG dataset. The architecture of each model, which includes generative adversarial networks (GANs), tabular variational autoencoders (TVAEs), and random sampling, is specifically selected based on the input features to ensure accurate simulation. This multimodel ensemble pipeline allows us to generate realistic synthetic donors for the trained tumor types. See also Figure S1.

of simulated mutations exactly match those in the training set, a lower rate than the 0.28% duplication of mutations among donors within each tumor type (Table S1). Finally, a VAF is assigned to each mutation (sampling model), resulting in a variant call format (VCF) file for each simulated donor.

At this stage, if specified by the user, the CNA-SV profile is used to simulate CNA (CTAB-GAN+ model 5) and SV (CTAB-GAN+ model 6) events. The assembly process for CNAs and SVs proceeds as follows. First, aberrant segments are generated based on the number of CNA events generated for a specific donor and tumor type, and their total length is adjusted to match the total simulated length of aberrant CNAs. Normal segments are then generated to fill the remaining genome length. Both aberrant and normal segments are shuffled and their sizes

adjusted to ensure they remain within chromosomal boundaries. Then, driver CNA events are randomly sampled based on the distribution observed for each specific tumor type and added to the simulated CNAs, replacing any overlapping non-driver segments and adjusting their boundaries to the surrounding alterations. Following the generation of CNAs, deletion and duplication SV breakpoints are automatically assigned based on the copy number profile, ensuring that all segments have their corresponding breakpoints. Finally, other SVs, such as inversions and translocations, are simulated and randomly assigned according to the frequent positions of SVs observed in cancer genomes (TVAE model 2), following an approach similar to the one used for point mutations (see STAR Methods). SVs that fall within homozygous deletions are excluded.

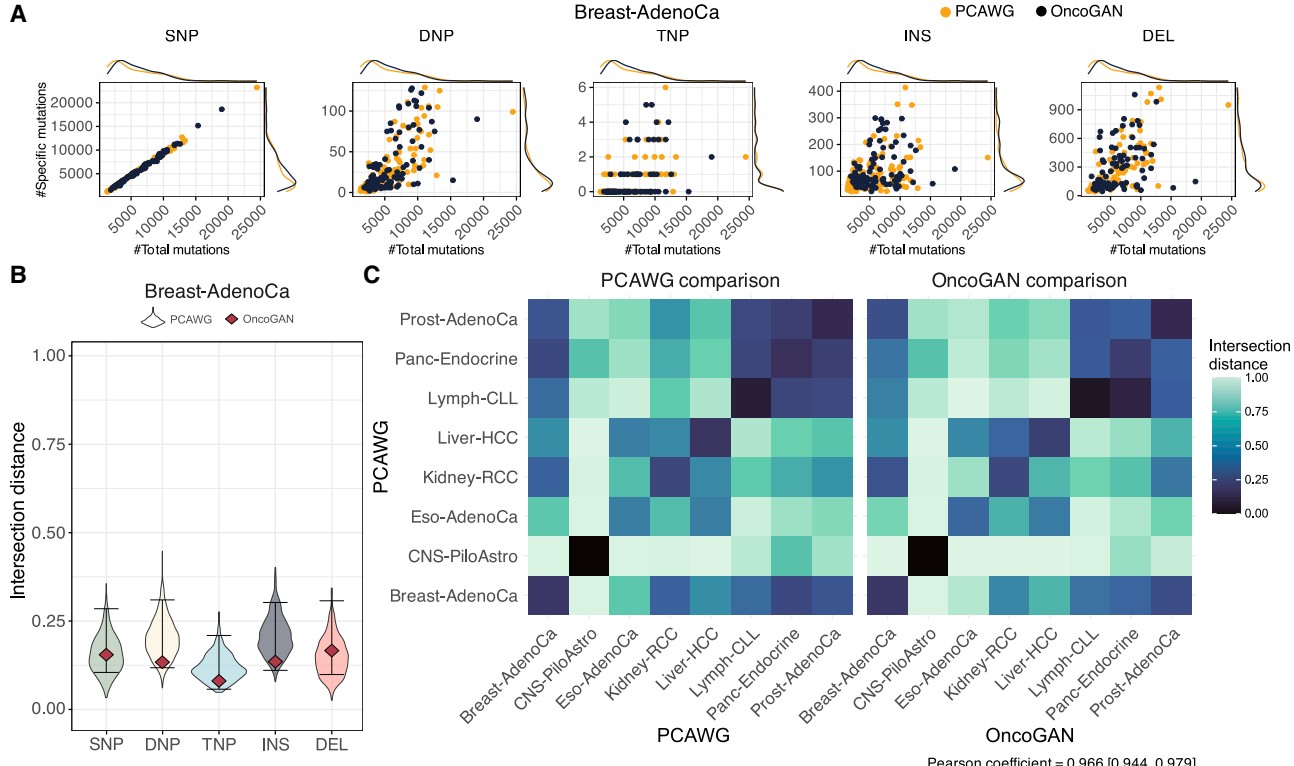

**Figure 2. Quality control plots for donor characteristics**

(A) Scatter and density plots comparing the number of specific mutation types to the total number of mutations for each donor (real in orange and simulated in black) in the Breast-AdenoCa dataset. PCAWG and OncoGAN distributions are highly similar.

(B) Violin plots for the Breast-AdenoCa tumor type, showing the distribution of intersection distances between two randomly sampled populations from the PCAWG dataset (1,000 iterations) and the scores comparing OncoGAN simulations to the actual dataset. For Breast-AdenoCa, all the mutation types have an intersection distance similar to that found in the real data.

(C) Heatmaps of intersection distances comparing different tumors from the PCAWG dataset (left) and OncoGAN simulations (right). A Pearson correlation of 0.966 indicates a high degree of similarity in the relationship between simulated and real donors. A lower intersection distance indicates greater similarity between the two populations.

Error bars on the violin plots represent confidence intervals calculated via bootstrapping the OncoGAN dataset (1,000 iterations). See also Figures S2 and S3.
SNP/DNP/TNP, single/double/triple nucleotide polymorphisms; INS, insertions; DEL, deletions.

After CNAs and SVs are generated, a plot is produced for each donor, displaying the CN profile alongside the positions of SV events. The CNAs and SVs are saved in the same format used by the PCAWG project. Additionally, the system generates a file containing information about the order and alleles in which events (mutations and CNA-SV events) occurred in the tumor, enabling accurate FASTA or BAM generation for downstream tasks.

### OncoGAN mimics tumor heterogeneity and clonality

To evaluate whether OncoGAN can simulate donors with characteristics that reflect the tumor heterogeneity observed among patients, we generated 100 samples for each of the eight tumor types and compared their characteristics, including the number and types of mutations, their VAF, and the relationships between driver genes.

#### Mutation density

First, we visually examined the distribution of the number of mutations in both real and simulated donors (Figure 2A; Figure S2).

This involved comparing the number of each mutation type against the total number of mutations per donor, allowing us to assess not only the range of specific mutations but also their proportion relative to the total mutation count. The resulting scatter and density distributions for the real and simulated data were highly similar, meaning that we were simulating heterogeneous donors. To quantify this similarity, we measured the intersection distance between real and synthetic population histograms for each type of mutation (Figure 2B; Figure S3; Table S2), where a smaller distance indicates greater similarity. For all simulations, except for certain mutation types in Liver-HCC and Prost-AdenoCa, the results were comparable to those observed in comparisons between subpopulations from the PCAWG dataset, suggesting that the simulated and real populations are indistinguishable. In the case of Liver-HCC, the differences in intersection distances may be attributed to its large sample size, which makes donors with very low or high mutation counts rarer and more challenging to simulate. Nonetheless, the scatter and density plots (Figure S2, Liver-HCC) show that the simulated

populations are quite similar. For Prost-AdenoCa, the main difference lies in the simulation of TNPs, likely attributable to the very low frequency of this mutation type, which represents only 0.006% of the total number of mutations, meaning that the differences are minimal. There is also a minor discrepancy in the simulation of deletions. However, the scatter and density plots show no perceptible differences between the two populations (Figure S2, Prost-AdenoCa). We also examined the similarity between different tumor types in terms of total mutation counts to determine if OncoGAN simulations exhibit the same patterns as real tumors. To do this, we compared the intersection distances of the different tumors from the PCAWG dataset and our simulations. As shown in Figure 2C, the relationships among tumor types in both cases are nearly identical (Pearson coefficient = 0.966 [0.944, 0.979]).

### Variant allele frequency

Another characteristic simulated by OncoGAN is the donor VAF. We implemented two approaches: a simple distribution, used when CNAs and SVs are not simulated, and a CNA-driven distribution.

The first approach reflects sample purity and subclonal heterogeneity, making it suitable for cases where CNAs and SVs are not simulated or for applications such as benchmarking variant callers—since it includes a range of clonal and subclonal VAFs—or for mutational signature discovery, where VAF is not considered. This is generated using a simple approach where the donor VAF is sampled from the real distribution, and mutation-specific VAFs are subsequently sampled according to the distribution for that particular donor VAF. As seen in Figure S4 and Table S3, there are no differences between PCAWG donor VAFs and our simulations, except for CNS-PiloAstro (Wilcoxon test $p = 0.008$, OncoGAN [$n = 100$]-PCAWG [$n = 89$]). However, the median VAF was 0.222 (q1 = 0.196; q3 = 0.260) and 0.238 (q1 = 0.221; q3 = 0.257) for PCAWG and OncoGAN donors, respectively, with identical variance (F score = 0.992 and F test $p = 0.964$ for CNS-PiloAstro; OncoGAN [$n = 100$]-PCAWG [$n = 89$]), overall indicating that both populations were very similar. When examining VAF for individual mutation types, the distribution is also quite similar between real and simulated data, with two notable differences (Figure S5). TNPs tend to have a lower VAF in the original dataset, likely due to the small number of TNPs present, while Lymph-CLL indels (insertions and deletions) show a higher overall VAF, possibly resulting from variant callers being less effective at detecting indels at lower frequencies.

To adjust mutation allele frequencies according to CNAs, we adopted a sequential simulation approach that randomizes the order of mutation and CNA events in the tumor genome. To validate this approach, we simulated germline mutations for one donor and assigned their VAFs accordingly. As shown in Figure S6, the VAFs of germline mutations reflect the underlying CNAs. Somatic mutations follow a similar pattern, with their VAFs depending on whether they occurred before or after the CNA. In this model, diploid regions have a mean VAF of 0.5; however, due to tumor purity, VAFs in solid tumors are typically lower. To account for this, we implemented a "normal-in-tumor" option. A practical example of this is shown in Figure S7B, where we used OncoGAN outputs and the InSilicoSeq tool implementation to generate a synthetic sequencing alignment file (BAM)

in which both germline and somatic mutations are shaped by the underlying CNAs.

### Driver mutations

The mutations simulated by our models are randomly generated to mimic the background mutational patterns observed in real tumors, and their effects on fitness are not taken into account; hence, their frequency distributions reflect those of passenger mutations rather than drivers. To model driver mutations, we sample them from a list specific to each tumor type published by the PCAWG. However, because driver mutations do not appear at the same frequency and some are mutually exclusive, we trained a model to replicate coding and non-coding driver mutation frequencies and their intercorrelation. After this simulation, the correlation between the frequency and the interrelationships of real and simulated driver mutations was very high, with a Pearson coefficient above 0.9 for most of the tumor types (Figure S8; Table S4). The tumor with the lowest correlation was Eso-AdenoCa (Pearson coefficient = 0.767 [0.759, 0.776]), likely due to the large number of selected drivers for this tumor (10 coding and 60 non-coding), making it more challenging to train an accurate model given the number of available donors.

## Synthetic tumors resemble tissue-specific mutational patterns

### Genomic distribution

We next sought to confirm that the mutational patterns specific to each type of cancer were accurately generated by our model. While the majority of mutations are passengers, their genomic locations are tumor-type specific and depend on the cell of origin.[7,32,33] To create high-quality synthetic cancer genomes, it is essential to replicate these specific mutation patterns along the genome. We compared the percentage of mutations found across the genome (in 1 Mb regions) between the PCAWG and the OncoGAN datasets, demonstrating that both real and simulated patterns are very similar (Pearson coefficient for Breast-AdenoCa dataset = 0.878 [0.869, 0.886]), with minor differences in some specific regions (Figure 3A; Figure S9). A clear example of this accuracy is observed in Lymph-CLL, where we successfully reproduced the mutational peaks associated with somatic hypermutation of the immunoglobulin genes. Moreover, the t-SNE visualization (Figure 3B) shows that PCAWG and OncoGAN donors cluster in very similar ways, easily distinguishing the different tumor types. This suggests that OncoGAN is accurately capturing the genomic positional patterns. The most notable discrepancies were observed in the Liver-HCC cancer type. To investigate the source of this discrepancy in the genomic distribution of mutations, we analyzed the variance in mutational density across real and simulated donors. As shown in Figures S10 and S11, there are a few very specific regions where our models failed to reproduce the observed variance. In these cases, instead of simulating donors with either a high or a low number of mutations in those regions, the model generated an average number of mutations. Nevertheless, as shown in Figure S9, the overall genomic distribution pattern for Liver-HCC is well captured.

A similar consistency is also observed when comparing the percentage of mutations detected per region in both datasets ($R^2$ for PCAWG = 0.679 [0.638, 0.709] and $R^2$ for

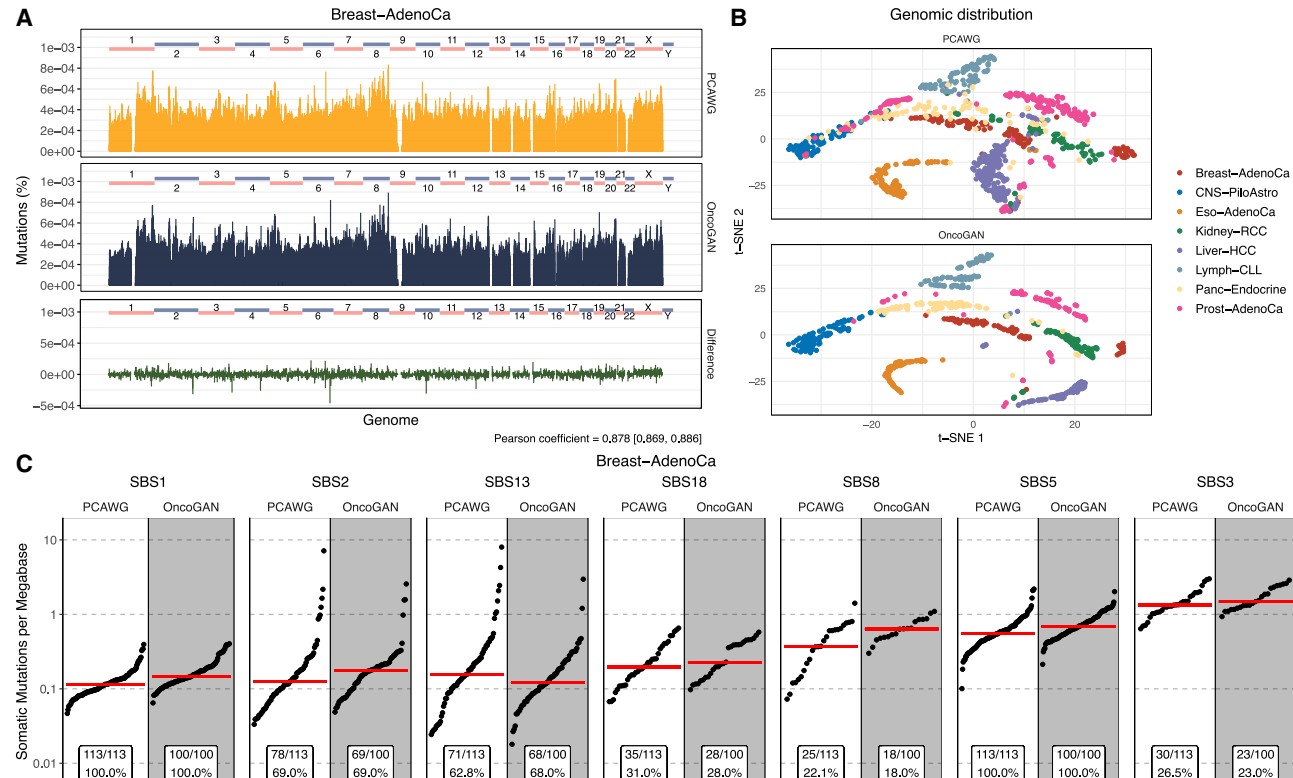

**Figure 3. Quality control plots for tissue-specific mutational patterns for Breast-AdenoCa tumors**
(A) Histogram displaying the total percentage of mutations across the genome in 1 Mb bins. Real and simulated donors are displayed in orange and black, respectively, and the difference is shown in green. A Pearson correlation of 0.878 indicates high similarity between PCAWG and OncoGAN results.
(B) t-SNE visualization of donors' genomic distribution (1 Mb windows), colored by tumor type, for both the PCAWG and the OncoGAN datasets.
(C) Distribution of mutation signatures, showing the number of somatic mutations per megabase and the percentage of donors exhibiting each signature. Signatures in the PCAWG dataset are accurately reproduced by OncoGAN, matching both the proportion of affected patients and the mutation density per megabase. Dots represent individual donors; the red line illustrates the mean number of somatic mutations per megabase.
See also Figures S9–S15 and S19. SBS, single base substitution.

OncoGAN = 0.772 for Breast-AdenoCa) (Figure S12; Table S5). The tumor type with the lowest similarity between real and simulated genomic coordinates is CNS-PiloAstro, with a Pearson coefficient of 0.683 and an $R^2$ of 0.466. This is comparable to the similarity observed when comparing two samples from the PCAWG dataset, which yields an $R^2$ of 0.276 [0.247, 0.301]. These values, which are low in comparison to those of other tumors with a higher mean number of mutations—such as Eso-AdenoCa ($R^2$ = 0.977 [0.969, 0.982]), Liver-HCC ($R^2$ = 0.983 [0.980, 0.985]), or Kidney-RCC ($R^2$ = 0.967 [0.946, 0.982])—can be attributed to the low mutational burden of CNS-PiloAstro (Table S5).

**Mutational signatures**
Cancers have distinct mutational signatures related to different biological or external mechanisms, and the mixture of signatures observed in any particular cancer depends on its tumor type, its treatment history, and the donor's environmental exposures. For example, SBS4 is associated with tobacco smoking, SBS7 with ultraviolet light exposure, and SBS9 with the somatic hypermutation activity of the activation-induced cytidine deaminase (AID) in lymphoid cells.[7] To accurately replicate these signatures, we need to simulate the frequency of donors with each

signature for a specific tumor type, the density at which each signature is found in each donor, and the relationships among different signatures (e.g., some are mutually exclusive). Thus, we predict the number of mutations for each mutational signature concurrent with the number of specific mutation types.

To test the fidelity of the mutation type distributions produced by OncoGAN, we first compared the distribution of mutation types for each tumor type in synthetic vs. real data, demonstrating that our model successfully simulates the intertumor heterogeneity among signatures (Figure S13). As a further test, we applied SigProfilerExtractor[34] to reconstruct the mutational signatures in our simulated genomes and compare them to the expected distribution of signatures for each tumor type (Figure 3C; Figure S14). We found the SigProfiler mutational distribution profiles derived from the synthetic data to be highly similar to that observed in real donors, with our simulations accurately capturing the variability among donors.

The particular signatures identified by SigProfiler in the OncoGAN-generated genomes matched the signatures identified in real genomes in almost all cases. We were able to identify all expected signatures across six of the eight modeled tumor types. The exceptions were Eso-AdenoCa (SBS8 and SBS10c)

and Panc-Endocrine (SBS19 and SBS30), where two signatures were missing in each type. This limitation is related to the technique we are using. Low-frequency mutational signatures, like SBS10c in Eso-AdenoCa and SBS19 in Panc-Endocrine, are underrepresented in the training dataset (5% of donors) and belong to complex tumor types with 8 and 10 different mutational signatures, making it more challenging for the model to learn the context distribution among the remaining mutations. In contrast, the SBS30 mutational signature is present in nearly 30% of the original donors. A more in-depth analysis revealed that its primary mutational context (C>T) was being simulated according to its distribution in the training data (Pearson coefficient = 0.986 [0.96, 0.995]). However, when the training data were compared to the reference profile available in COSMIC and used by SigProfiler, it showed the second-lowest correlation, with a Pearson coefficient of 0.798 [0.515, 0.924]. The lowest correlation was observed for SBS19, which was the other missing signature, with a coefficient of 0.577 [0.038, 0.856]. In contrast, the average correlation for the rest of the detected signatures in Panc-Endocrine was 0.886 (Figure S15; Table S6).

To further assess the accuracy of the generated signatures, we compared the number of specific mutational signatures simulated by OncoGAN with those detected by SigProfiler for each donor. We found that, for more than 90% of the donors, these two numbers were highly correlated (Pearson coefficient from 0.827 [Liver-HCC] to 0.997 [Kidney-RCC]) (Figure S16; Tables S7 and S8). OncoGAN simulated a specific signature that was not detected by SigProfiler in only 7.2% of cases. However, in those cases, the simulated signature appeared at a very low frequency. On average, signatures detected by SigProfiler had 2,490 simulated mutations, while those not detected had only 1,342 mutations (Table S7). This suggests that these donors may fall below the tool's detection limit.

### Indels

Another feature we are simulating, along with the mutational context, is the length of the indels. We compared the indel length distributions in the PCAWG and OncoGAN datasets and found that they are within the same range, following the same pattern (Figure S17). There are some statistical differences for certain indel lengths and tumor types due to sample size, but these do not significantly affect the quality of the simulation.

### Mutation consequences

Finally, we explored the consequences of the mutations in the genome using the Ensembl Variant Effect Predictor tool,[35] comparing the results between real and simulated donors. As shown in Figure S18, for each of the tumor types, the proportions of mutations per donor assigned to the possible consequences are highly similar, with comparable variability across both datasets. Intergenic and intronic variants are the most common categories in all cases, followed by downstream and upstream variants. On average, missense mutations account for 0.64% and 0.66% of the total number of mutations in PCAWG and OncoGAN donors, respectively.

### Performance of driver calling algorithms in driver cancer datasets

As previously explained, the mutations generated randomly by OncoGAN are passenger mutations. However, as we have

demonstrated, we introduced driver mutations maintaining the same frequency and interrelationships observed in real patients. To validate whether the same driver genes can be detected in our synthetic datasets, we applied the widely used ActiveDriverWGS[36] algorithm to identify driver genes in both the PCAWG and the OncoGAN datasets. The performance of ActiveDriverWGS was highly accurate for both the PCAWG and the OncoGAN datasets, detecting 89% and 87%, respectively, of the coding genes for which driver mutations were considered across the eight studied tumor types (Table S9). When we examined the driver genes that were not identified by ActiveDriverWGS, we found that these were genes mutated at very low frequencies (<10% donors) in the simulated and real datasets.

### Performance of tissue-of-origin prediction

Ensuring that simulated genomes not only replicate individual characteristics but also maintain tumor-type specificity is essential for validating their biological relevance. As we have shown previously, the individual characteristics OncoGAN simulates closely resemble the original data. To assess whether the final simulation (i.e., each individual simulated donor) corresponds to the expected tumor type, we utilized DeepTumour, a tool that predicts the tumor tissue of origin based on the list of detected somatic mutations.[37] We ran DeepTumour on the synthetic donors and achieved very high prediction accuracy of nearly 100% for most of the tumor types (Table S10). However, when examining the 5-fold cross-validation results of the original model, we observed that some tumor types had lower performance (Figure 4C, baseline metrics). In particular, we investigated the Lymph-MCLL and Eso-AdenoCa datasets in more detail to understand the drop in performance for these tumor types.

Lymph-CLL is a disease characterized by two distinct subtypes, unmutated (U-CLL) and mutated (M-CLL), based on the mutational status of the immunoglobulin heavy-chain variable region (IGHV), with M-CLL presenting three mutation peaks at the IGHV genes (chr2-IGK, chr14-IGH, and chr22-IGL, Figure S9) and the mutational signature SBS9, while U-CLL lacks these features.[37] When analyzing the 5-fold cross-validation results from DeepTumour, we observed that the recall for Lymph-MCLL was the lowest, with only 63% (22/35) of donors correctly classified, and the remaining 37% (13/35) misclassified as B cell non-Hodgkin's lymphoma (Lymph-BNHL) (Table S10). This misclassification can be attributed to the nearly identical mutational patterns between these two tumors, with a Pearson coefficient of 0.941 and $R^2$ of 0.887 (Figure S19). The primary difference occurs in the three regions that correspond with the IGHV genes, which have a lower percentage of mutations, as only 73% of Lymph-BNHL cases present the SBS9 signature, compared to 100% of M-CLL donors. Furthermore, of the 95 Lymph-CLL donors used to train DeepTumour, only 35 (37%) were M-CLL, indicating that a larger sample size may be necessary to improve the prediction accuracy for this tumor type.

On the other hand, Eso-AdenoCa had the lowest precision (76%) and the second lowest recall (80%) among the tumor types. To investigate the Eso-AdenoCa donors, we performed a principal-component analysis (PCA) based on the number of

**CellPress**

**Cell Genomics**
Article

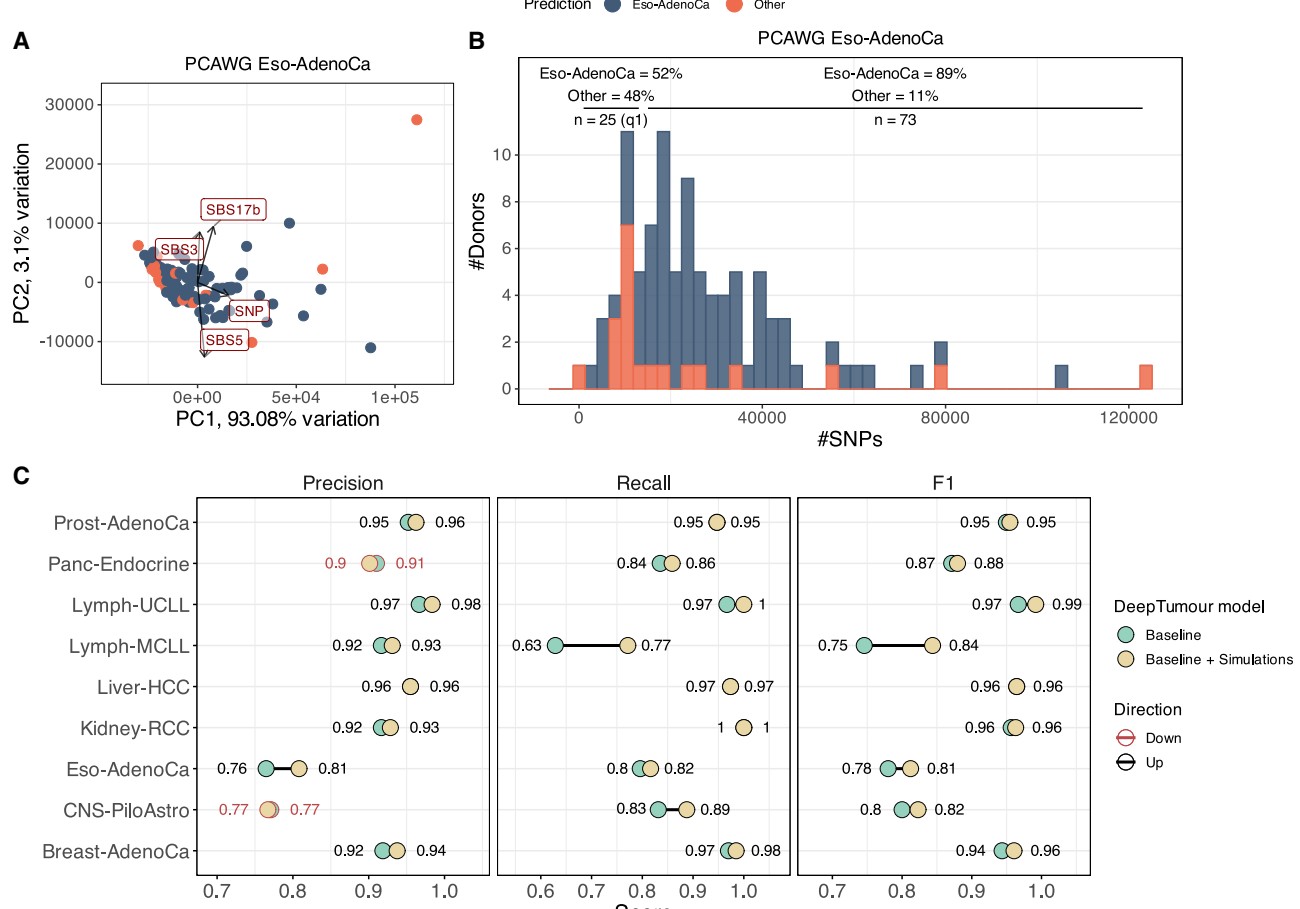

**Figure 4. DeepTumour prediction analysis**
(A) PCA of Eso-AdenoCa real donors using the total number of each mutation type and mutational signatures as features. The four most significant features are highlighted in red boxes.
(B) Histogram showing the distribution of accurate and misclassified Eso-AdenoCa donors by DeepTumour according to the number of SNPs. Donors with fewer mutations are more frequently misclassified (48% vs. 11%).
(C) Comparison of precision, recall, and F1 metrics between the baseline DeepTumour model (light green) and a model trained with the original dataset plus 100 simulated donors for each of the eight tumor types (beige). Metrics that decreased in the new model compared to the baseline are marked in red text.
See also Figure S20.

each type of mutation and signature. As shown in Figure 4A, misclassified donors can be mainly distinguished from the rest by their number of SNPs. Specifically, when dividing the donors by their total number of SNPs, we found that those with the fewest mutations (q1, *n* = 25) were more frequently misclassified (48% vs. 11%) (Figure 4B). This suggests that these samples may represent cases that are underrepresented in the training dataset.

### Improving algorithms' performance by combining real and synthetic datasets

Given these findings, we decided to train a new DeepTumour model using a mixed dataset of real and synthetic donors to determine whether this could enhance the model's accuracy. We trained DeepTumour with the same 5-fold cross-validation strategy, but adding 100 OncoGAN-simulated donors to 8 of 29

tumor types present in the training dataset. For this new model, the overall accuracy improved by 0.9% (from 89.26% to 90.16%) (Figure S20). For the eight studied tumor types, the 5-fold cross-validation precision, recall, and F1 scores increased by 1%, 1.58%, and 1.34%, respectively. Additionally, tumor types with fewer donors in the original DeepTumour training dataset benefited the most from the supplementary training, for example, Lymph-MCLL (*n* = 35) and Lymph-UCLL (*n* = 60), with F1 score increases from 75% to 84% and from 97% to 99%, respectively (Figure 4C). On the other hand, two tumor types experienced a drop in one of the metrics, CNS-PiloAstro (−0.4% precision) and Panc-Endocrine (−0.9% precision), likely due to an increase in Thy-AdenoCa donors misclassified as CNS-PiloAstro, and CNS-Oligo donors misclassified as Panc-Endocrine, possibly due to the low number of donors available for those tumor types (*n* = 48 and *n* = 18, respectively) (Figure S20).

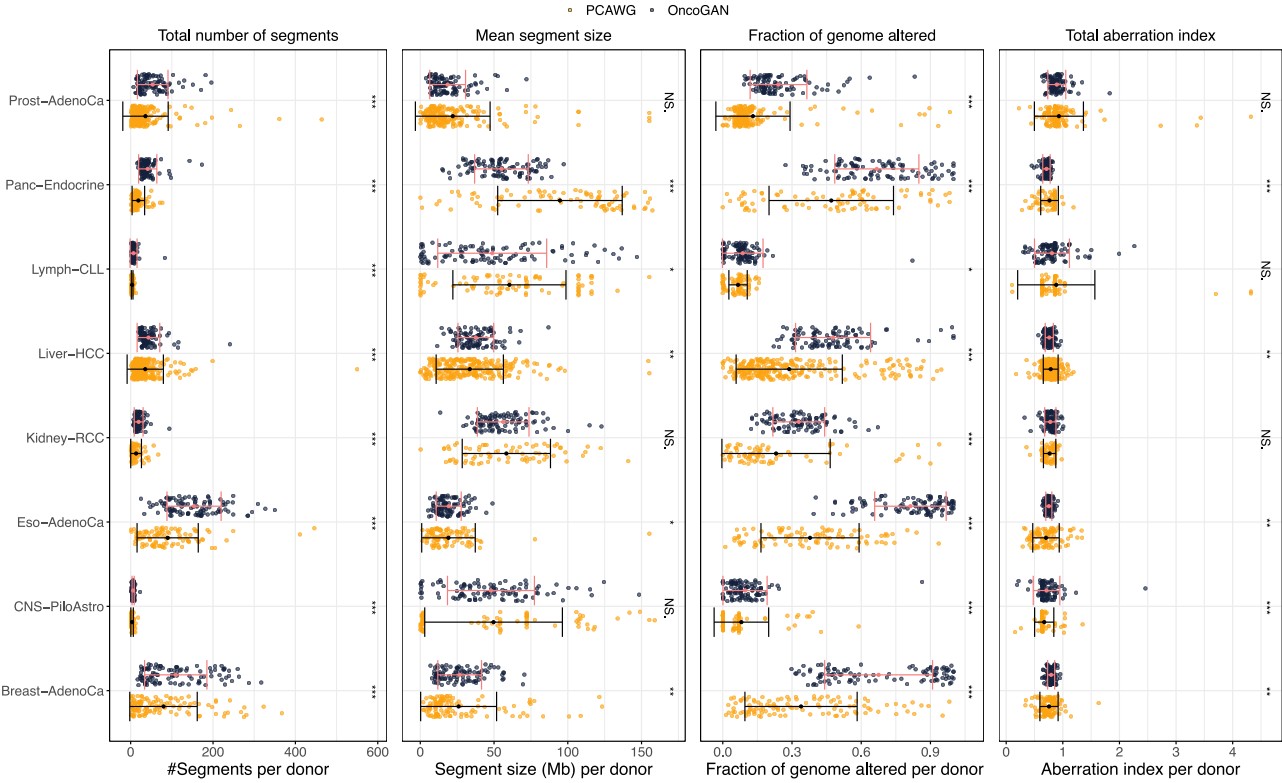

**Chromosomal instability scores**

· PCAWG  · OncoGAN

**Figure 5. Chromosomal instability scores to measure the similarity in copy number alterations between real (orange) and simulated (black) donors, demonstrating high concordance between the two groups**

The total number of segments indicates the number of altered regions per donor (i.e., regions where the major and minor alleles differ from 1). The mean segment size refers to the average length of the altered regions. The fraction of the genome altered is calculated by summing the lengths of all altered segments and normalizing by the total genome size. The total aberration index (TAI) measures the average deviation from the normal copy number state. The sample size used for each comparison corresponds to the number of donors available for each tumor type. Error bars show the standard deviation of the dataset. The Wilcoxon test was used to compare the groups. NS, $p > 0.05$; *$p \leq 0.05$, **$p \leq 0.01$, and ***$p \leq 0.001$.

See also Figure S21.

## Expanding the simulation to copy number alterations and structural variants

To simulate an accurate representation of CNAs and SVs for each donor, we trained a model that uses the total number of point mutations as the basis for generating these alterations (Figure S21). For CNAs, we defined segments of varying lengths and simulated the minimum and maximum numbers of alleles for each segment, ensuring these features followed the same distributions observed in real patients. To ensure that potential CNA driver events were also represented, we randomly selected alterations from the most frequently altered regions for each tumor type, based on their observed frequencies in real data (see STAR Methods). To measure the quality of the simulations we calculated several chromosomal instability scores (CISs), such as the total number of segments and their mean size, the fraction of the genome that is altered (FGA), or the total aberration index (TAI) (Figure 5).[38] The simulated tumor types show high concordance with the PCAWG dataset, with the exception of Breast-AdenoCa and Eso-AdenoCa, which differ in the FGA metric due to the simulation of a higher number of altered segments. For SVs, duplication and deletion events were automati-

cally assigned based on copy number status, while the number of inversions and translocations was simulated to match the distributions observed in real donors. As shown in Figure S22, the numbers and lengths of these events between OncoGAN and PCAWG donors for each tumor type are highly similar. In the case of translocations, PCAWG does not report the actual size of each event, since for most translocations only a single breakpoint is detected. Although the comparison between real and simulated donors is statistically significant in many cases (Wilcoxon test $p < 0.05$), the differences between the two datasets for the eight different tumor types are small, as evidenced by the similarity of their means (Table S11).

## Compatibility with FASTQ and BAM generation tools

In addition to the direct applications of the simulated VCFs—such as analyzing mutational signatures, benchmarking driver gene detection, and augmenting data to improve existing algorithms—OncoGAN simulations can also be used to generate personalized BAM files for benchmarking genomic analysis tools that operate at the aligned read level. To illustrate OncoGAN's compatibility with FASTQ and BAM generation tools, we provide

a module to integrate OncoGAN-generated VCF and CNA-SV files with two widely used tools: InSilicoSeq[39] (v.2.0.1) and BAM-surgeon[19] (v.1.4.1).

For InSilicoSeq, we first create a custom FASTA file containing a set of user-provided common SNPs, along with the mutations and CNAs generated by OncoGAN, following the event order defined in the pipeline (see STAR Methods). This approach enables the simulation of a range of VAFs depending on whether the mutation occurs before, after, or between CNA events. If the user employs OncoGAN solely to simulate mutations, we take advantage of InSilicoSeq's capability to assign different frequencies to each entry in a multi-FASTA file. This allows us to create multiple copies of the same chromosome at varying frequencies, each spiked with different mutations according to the assigned VAF, thereby simulating a range of clonal and subclonal VAFs. In both approaches, we begin with two copies of each chromosome to represent the diploid normal genome, enabling the simulation of phased haplotypes (Figure S7). The resulting custom FASTA file can also be used directly with any other FASTA-to-FASTQ read generation tool.

For BAMsurgeon, we automatically convert the VCF and SV-CNA formats into the ones required by the tool. The user must provide their own template BAM file. In this case, the same command used to generate a custom FASTA using InSilicoSeq can also be employed to create a basic synthetic BAM containing only germline mutations, which can then serve as input for BAMsurgeon. Note that BAMsurgeon will create its own haplotypes, replacing the ones generated by OncoGAN. Detailed instructions on how to use OncoGAN with both InSilicoSeq and BAMsurgeon are available in our GitHub repository.

These approaches can be readily adapted to work with other FASTQ/BAM generation tools or extended to produce more complex outputs, enabling the creation of an unlimited number of realistic synthetic genomes.

### Open access repository of simulated cancer genomes

To provide a foundation for community cancer genome analysis benchmarking efforts, we have used the OncoGAN pipeline to generate 800 simulated cancer genome VCFs and matched CNA and SV files across a diverse set of eight tumor types. These simulations are now publicly available for download and redistribution through our project repositories on HuggingFace (https://huggingface.co/datasets/anderdnavarro/OncoGAN-syntheticVCFs) and Zenodo (https://doi.org/10.5281/zenodo.13946726), with additional tumor types and features to be included in future releases as we continue to train and refine the models. Because VCF is a widely accepted standard for mutation calling files and the formats used for CNAs and SVs are the same ones reported in the PCAWG study, these datasets can be easily integrated into the existing range of bioinformatic tools, facilitating their adoption by researchers in the field.

### DISCUSSION

Here, we present OncoGAN, a generative AI pipeline that leverages deep learning algorithms to simulate highly realistic synthetic cancer genomes. We have demonstrated that OncoGAN reproduces features found in various tumor types with a high level of fidelity while also capturing the significant heterogeneity found across donor populations. By applying the widely used cancer genome analysis tools SigProfilerExtractor,[34] ActiveDriverWGS,[36] and DeepTumour,[37] we showed that the synthetic genomes produced by OncoGAN discovered the same mutational signatures, cell-of-origin signatures, and driver mutation profiles as they do when applied to real data. Finally, we showed that by supplementing real training-set data with OncoGAN-simulated data, we could improve the performance of the DeepTumour tumor-type prediction model, suggesting that synthetic genomes could be useful in augmenting the training sets of other genome machine learning applications. In this regard, we successfully worked with tumors that have a small number of donors, such as Lymph-MCLL and Lymph-UCLL, with 35 and 60 donors, respectively. This demonstrates that OncoGAN has the potential to augment datasets for other rare tumor types.

We believe that these highly realistic synthetic cancer genomes can help address some of the current challenges in genomics, such as data accessibility,[20–22] by promoting data sharing while protecting patient confidentiality.[40] We have demonstrated that OncoGAN effectively protects patient privacy by simulating the position and context of mutations independently. Additionally, the training set contains only somatic mutations (with no germline information) from which donor identifiers have been removed. As a result, our synthetic genomes can be made open access and fully available to the research community. This approach aligns with initiatives like the Common Infrastructure for National Cohorts in Europe, Canada, and Africa (CINECA)[41] or the work by D'Amico et al.,[42] who have synthesized cohort datasets containing phenotypic data. Moreover, data augmentation using synthetic genomes can help resolve class imbalance and small sample size issues,[40,43] providing valuable resources for training new algorithms or improving existing ones, as demonstrated in the DeepTumour example. This approach has also been successfully applied in other fields, including medical image analysis,[44] survival analysis,[45,46] and single-cell data.[47,48]

In recent years, several alternative approaches have been developed for the simulation of synthetic genomes, each employing one of three main strategies. BAMsurgeon,[19] Xome-Blender,[49] and SomatoSim[50] each use a real sequencing alignment file (BAM) as input, add the alterations, and output the modified BAM along with a list of the incorporated mutations. In contrast, insiM[18] takes a similar approach by using a BAM file as input, but it outputs paired reads in FASTQ format. Finally, simuG[51] and Mutation-Simulator[52] use the reference genome in FASTA format as input, producing an altered FASTA file, which can then be converted into FASTQ reads using any read simulator, such as InSilicoSeq[39] or Sandy.[53] However, all of these tools primarily focus on integrating random mutations and/or more complex alterations, which do not accurately represent real tumor characteristics. In comparison, OncoGAN is the only method capable of simulating realistic mutational and copy number profiles specific to different tumor types. Moreover, as we have demonstrated, it can be integrated with any of the previous approaches to generate more realistic in silico sequencing files, thereby enabling synthetic genomes to address more complex tasks.

Currently, OncoGAN can generate synthetic donors for eight tumor types, enabling a variety of downstream applications, including benchmarking and training of variant callers, discovery of mutational signatures, evaluation of tools for driver mutation and CNA detection, and data augmentation for specific tasks. In future work, we plan to expand OncoGAN to support additional tumor types, including those that are less common, to simulate a broader range of tumor characteristics—such as indel signatures, subclonal reconstruction, or population-based germline haplotypes—and to improve the compatibility with FASTQ/BAM generation tools in order to simulate complex SVs.

In conclusion, OncoGAN represents a significant advance in the generation of synthetic cancer genomes. It offers a solution to current challenges in data accessibility and privacy while also serving as a powerful tool for enhancing algorithm development and benchmarking.

### Limitations of the study

OncoGAN has certain limitations stemming from the factorization approach it employs. For example, it is currently unable to simulate subclonal reconstructions, as doing so would require modulating mutational signatures and genomic mutation density to realistically represent distinct subclonal populations. In this regard, other tools have demonstrated good accuracy in simulating tumors with subclonal architecture and can be used for this purpose.[49,54] Another limitation of the factorization approach is the lack of integration between driver mutations and CNAs; these features are currently modeled independently and linked only through the total number of mutations present in the simulated donor.

Additional limitations arise from the biological complexity of tumors and the scarcity of paired datasets needed to train the models for certain mechanisms. For example, our model assumes that the genomic distribution of mutations is influenced by chromatin accessibility—i.e., open chromatin regions are expected to show lower mutation density.[32] However, this assumption may be insufficient to fully capture the impact of chromatin state. Similar limitations apply to other biological features such as replication timing and strand bias, which may be partially inferred by the mutational density or signature models, but are not explicitly simulated. Regarding CNAs and SVs, our model does not support whole-cell complex events such as chromothripsis, chromoplexy, or chromoanasynthesis.

### RESOURCE AVAILABILITY

#### Lead contact

Requests for further information and resources should be directed to and will be fulfilled by the lead contact, Lincoln Stein (lincoln.stein@gmail.com).

#### Materials availability

This study did not generate new, unique reagents.

#### Data and code availability

- The OncoGAN software is publicly available on GitHub (https://github.com/LincolnSteinLab/oncoGAN) and DockerHub (https://hub.docker.com/r/oicr/oncogan) to enhance reproducibility.
- The training files, trained models, and simulated donors used in this paper (800 simple VCFs and 800 VCFs with their matched CNAs and SVs) have been uploaded to HuggingFace (https://huggingface.co/

collections/anderdnavarro/oncogan-67110940dcbafe5f1aa2d524) and Zenodo (https://doi.org/10.5281/zenodo.13946726).
- All scripts used to preprocess the raw files, prepare the training files, analyze the results, and generate all figures and tables can be found in the Source Data folder on Zenodo (https://doi.org/10.5281/zenodo.13946726).

### ACKNOWLEDGMENTS

We thank Matus Medo, Quaid Morris, Valli Subasri, and Phil Fradkin for their useful suggestions. This work was performed using high-performance computing resources from the Ontario Institute for Cancer Research. A.D.-N. was supported by the Ontario Genomics-CANSSI Ontario Postdoctoral Fellowship program in Genome Data Science. X.Z., W.J., and L.S. were supported by funding from the Province of Ontario, Canada.

### AUTHOR CONTRIBUTIONS

A.D.-N.: conceptualization (equal), data curation (lead), formal analysis (lead), investigation (lead), methodology (lead), project administration (equal), software (lead), visualization (lead), writing – original draft (lead), and writing – review & editing (equal). X.Z.: formal analysis (supporting), methodology (supporting), visualization (supporting), writing – original draft (supporting), and writing – review & editing (supporting). W.J.: formal analysis (supporting) and methodology (supporting). B.W.: funding acquisition (supporting), project administration (supporting), supervision (supporting), and writing – review & editing (supporting). L.S.: conceptualization (equal), funding acquisition (lead), project administration (equal), resources (lead), supervision (lead), and writing – review & editing (equal).

### DECLARATION OF INTERESTS

The authors declare no competing interests.

### STAR★METHODS

Detailed methods are provided in the online version of this paper and include the following:

- KEY RESOURCES TABLE
- METHOD DETAILS
  ○ Data preprocessing
  ○ Training OncoGAN models
  ○ Augmenting DeepTumour training data
  ○ Custom genome reconstruction
- QUANTIFICATION AND STATISTICAL ANALYSIS
  ○ OncoGAN individual quality control metrics
  ○ DeepTumour prediction

### SUPPLEMENTAL INFORMATION

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

## STAR★METHODS

### KEY RESOURCES TABLE

| REAGENT or RESOURCE | SOURCE | IDENTIFIER |
|---|---|---|
| **Deposited data** | | |
| Synthetic tumor genomes (VCFs) | This paper | Zenodo: https://doi.org/10.5281/zenodo.13946726<br>HuggingFace: anderdnavarro/OncoGAN-syntheticVCFs |
| OncoGAN models | This paper | Zenodo: https://doi.org/10.5281/zenodo.13946726<br>HuggingFace: anderdnavarro/OncoGAN |
| **Software and algorithms** | | |
| OncoGAN (v0.2.1) | This paper | GitHub: LincolnSteinLab/oncoGAN |
| CTAB-GAN+ | Zhao et al. [29] | GitHub: Team-TUD/CTAB-GAN-Plus |
| CTGAN (v1.4.0) | Xu et al. [27] | Webpage: https://docs.sdv.dev/sdv/single-table-data/modeling/synthesizers/ctgansynthesizer |
| TVAE (v1.4.0) | Xu et al. [27] | Webpage: https://docs.sdv.dev/sdv/single-table-data/modeling/synthesizers/tvaesynthesizer |
| SigProfilerExtractor (v1.1.21) | Islam et al. [34] | GitHub: AlexandrovLab/SigProfilerExtractor |
| ActiveDriverWGS (v1.2.1) | Zhu et al. [36] | GitHub: reimandlab/ActiveDriverWGSR |
| Variant Effect Predictor tool (v112) | McLaren et al. [35] | GitHub: Ensembl/ensembl-vep |
| DeepTumour | Jiao et al. [37] | GitHub: LincolnSteinLab/DeepTumour |
| InSilicoSeq (v2.0.1) | Gourlé et al. [39] | GitHub: HadrienG/InSilicoSeq |
| BAMsurgeon (v1.4.1) | Ewing et al. [19] | GitHub: adamewing/bamsurgeon |

### METHOD DETAILS

#### Data preprocessing

##### Point mutations

The original file used to extract all features for training the different models was the final_consensus_passonly.snv_mnv_indel.icgc.public.maf.gz (GRCh37) from the ICGC 25K data portal[4]. This file contains all final mutations detected by whole genome sequencing (WGS) for each donor across the 25 tumor types studied in the PCAWG. Twelve tumor types were discarded due to an insufficient number of samples for model training (<50 donors). From the remaining thirteen tumor types, we selected eight, ensuring a heterogeneous dataset based on factors such as the tumor mutational burden, the number of available donors and the tumor subtype. The selected studies were: Breast-AdenoCa (n=113), CNS-PiloAstro (n=89), Eso-AdenoCa (n=98), Kidney-RCC (n=74), Liver-HCC (n=264), Lymph-CLL (n=95), Panc-Endocrine (n=85) and Prost-AdenoCa (n=191). For each mutation, we retained the following information: study (tumor type), donor ID, chromosome, position, length, type of mutation, reference and alternative bases, trinucleotide context and variant allele frequency. Mutations were categorized into five groups: single nucleotide polymorphisms (SNPs; A > C), double nucleotide polymorphisms (DNPs; AA > CC), triple nucleotide polymorphisms (TNPs; AAA > CCC), insertions (INS; A > AA) and deletions (DEL; AA > A). Mutations without variant allele frequency were excluded.

To associate each SNP with its corresponding mutational signature (single base substitution; SBS) from the COSMIC catalogue (https://cancer.sanger.ac.uk/signatures/sbs/), individual variant call format files were extracted for each donor. SigProfilerExtractor[34] (v1.1.21) was then run, individually for each tumor type, using its default configuration except for the –maximum_signatures option, which was set to 8. The De_Novo_MutationType_Probabilities.txt file was used to sample the most probable signature each SNP belongs to, based on donor ID and mutational context. SBSs present in 5% or fewer (rounded) of donors were removed from the dataset, as more data would be required to accurately train the models.

Since categorical columns in tabular data are converted into one-hot encoding, we simplified the dataset by removing the chromosome column. Thus, positions were transformed into a continuous range along the genome (e.g., if chromosome 1's length is 1000, position 1 on chromosome 2 would be 1001).

The preprocessed file was then used to prepare the training files for all the models described below except for CNA and SV related models.

##### Copy number alterations and structural variants

The training files for CNA and SV simulations were created using the original datasets available in the consensus.20170119.somatic.cna.icgc.public and final_consensus_sv_bedpe_passonly.icgc.public folders, respectively, from the ICGC 25K data portal[4]. Due to the limited number of events, all tumor types with more than 30 donors were included to ensure sufficient data for training the models.

While all donors were used, the tumor type of origin was retained as a feature in the training file for CNA-related events, enabling tumor-specific alteration simulations. However, for SV-related events, this feature was excluded due to the insufficient number of alterations within each tumor type. The lengths of the events were calculated using their start and end positions and then transformed using natural logarithms to reduce variability and improve data distribution. These processed datasets were used to generate four distinct training files, which are described below.

### Training OncoGAN models

Detailed hyperparameters settings for all the models can be found in Table S12.

#### Donor characteristics

Donor characteristics for point mutations refer to the number and types of mutations and signatures present in each donor. To generate this dataset, we counted the number of each mutation type per donor, replacing SNPs with the assigned SBSs to avoid duplicating information in the training file. We trained a separate model for each of the tumor types studied, using CTAB-GAN+[29] with 720 different hyperparameter combinations (epochs: 100 to 500 in increments of 20; batch size: 10 to 30 in increments of 5; learning rate: 0.001 to 0.01 in increments of 0.001). The best model or models were selected based on the similarity of scatter and density plots comparing the simulations to the original data. Given the complexity of tumors –where different mutational types and signatures interact in varied ways– and the limited data available for certain tumor types, we refined the simulations by combining models trained with different hyperparameter settings and applying tumor-specific ad-hoc filters. This approach allowed us to better capture low-frequency mutational signature patterns, ensuring that the generated data accurately resembled the original distributions.

For copy number alterations and structural variants, donor characteristics include features such as the tumor type, the number of CNAs and each type of SV (deletions, duplications, inversions, and translocations), the total length of aberrant CNA segments, and the total number of mutations per donor. The total number of mutations is used to guide the simulation of CNAs and SVs, ensuring they match the donor's mutational profile. Given the simplicity of the data, we trained a single CTAB-GAN+ model to simulate donor characteristics for CNAs and SVs, then the tumor type feature can be used to specifically simulate samples for the desired tumor.

#### Driver genes and CNAs

Since OncoGAN randomly generates positions and trinucleotide contexts, the resulting mutations represent only passenger mutations. Therefore, OncoGAN cannot directly simulate driver mutations, as it lacks the biological information necessary for this task. To address this limitation, we first selected a list of driver genes published by PCAWG and available at the ICGC 25K data portal (TableS1_compendium_mutational_drivers.xlsx). We classified mutations in driver genes as either "coding" or "non-coding," creating a new ID (e.g., TP53_coding / TP53_intron). Driver IDs were filtered to include only those present in at least 5% of donors for coding mutations and 10% for non-coding mutations. The remaining mutations were used to create a database from which mutations could be drawn during the simulation.

However, driver IDs cannot be randomly selected, as some drivers are more common than others, and some tend to occur together or be mutually exclusive. To simulate these driver's intercorrelation, we trained a specific model. The training file for this model consisted of a count of how many driver mutations each donor had for each driver ID.

The same approach used for training donor characteristics models was applied here, with one model trained for each tumor type (CTAB-GAN+[29] with 720 hyperparameter combinations). The best models were manually selected based on the frequency of mutations for each driver ID and the correlation between driver's intercorrelations.

In a similar way, OncoGAN cannot simulate CNA driver events. In this case, we did not have enough data to train a model capable of capturing CNA driver intercorrelations, as described above for driver mutations. Instead, we used the copy number states of cytobands from the PCAWG dataset (all_thresholded.by_genes.rmcnv.pt_170207.txt). CNAs were aggregated by tumor type, and the frequency of each event was calculated. Events occurring in fewer than 5% of donors were excluded from the final list.

#### Variant allele frequency

Two approaches have been implemented to simulate variant allele frequency. The first is a straightforward method that reflects tumor purity and is independent of the donor- and mutation-specific characteristics described above. First, we calculated the mean VAF for each donor and ranked them into 100 VAF bins (1% increments, from 0 to 1) to determine the percentage of donors that fall into each bin, independently for each tumor type. Then, within each of these ranks, we selected the mutations belonging to donors within each bin and further ranked them according to 40 VAF bins (2.5% VAF increments, from 0 to 1), to calculate the percentage of mutations in each bin. To simulate a donor, we first sample a mean donor VAF from the first file using the calculated probabilities and then sample mutation VAFs using the probabilities calculated in the second file for that specific mean donor VAF.

The second approach simulates allele frequencies based on copy number events, following a sequential order after shuffling mutations with CNA and SV events (deletions and duplications). After generating the list of somatic mutations and CNA events, each mutation is assigned to its corresponding CNA context. Duplication events are then repeated according to the difference between the observed and normal copy number (e.g., if a region has major_cn = 3 and minor_cn = 1, the major allele duplication will appear twice in the event list). Mutations and events are subsequently shuffled to simulate a random sequential order of appearance in the tumor genome. Following this order, alleles and their associated mutations are duplicated or deleted, and new mutations may be introduced in any of the available alleles at a given point. Finally, the VAF is calculated based on the number of mutated alleles and the total number of alleles at each position, starting from a normal distribution (mean = 0.9, SD = 0.15) to incorporate sequencing noise.

### Genomic position

The genomic position of mutations is a variable with a very wide range, spanning from 1 to 3 billion base pairs (3Gbp), making it difficult for generative models to capture this variability. To address this, we performed a two-step discretization of the genome. First, we compressed the 3Gbp genome into 30Mbp (a 1:100 ratio) to reduce the complexity, as single-base resolution is not necessary for passenger mutations. Next, we divided the compressed genome into 105 bins of 0.3Mbp each, matching each mutation's position to its corresponding bin, discarding bins with fewer than two mutations. We then used the CTGAN[27] model to simulate new bins at the same frequency found in the original data. In the second step, we trained a small tabular variational autoencoder (TVAE)[27] model independently for each bin, using the positions rather than the bins as input for the training. Finally, the simulated positions were expanded to the actual genome size. The use of both architectures is necessary because the GAN model better captures the overall frequency of the bins, but it is unable to accurately reproduce the specific mutation positions when working with a larger range.

The same strategy was used to train a model to simulate the position of the structural variants.

### Mutational signatures

To simulate the mutations, we prepared a training file containing the following features: position, length, VAF, reference and alternative contexts, and signature. The context is represented with individual bases in separate columns (r.ctx1, r.ctx2, r.ctx3, a.ctx1, a.ctx2, a.ctx3). Although we trained specific models for genomic position and VAF, we included these features in this training file as well, as they stabilized and improved model performance, preventing exploding issues in the gradient penalty during the GAN model's training phase.

We used a CTAB-GAN+[29] architecture for these models, defining the variables as follows: 'r.ctx1', 'r.ctx2', 'r.ctx3', 'a.ctx1', 'a.ctx2', 'a.ctx3', and 'signature' as categorical; length as a mixed type (with 0 representing "no indel" as a categorical value, >0 representing insertion size and <0 for deletion size); position as an integer; and VAF as a default float. Hyperparameters were selected based on tumor characteristics, with individual optimization for each tumor type. Batch size was set to the mean number of mutations per donor, the learning rate to 2e-4, and epochs ranged from 280 to 3200 to adapt the training time to an average of five days. The 'signature' variable was not simulated directly by the GAN model but was instead predicted by the auxiliary component of the CTAB-GAN+ model, based on the other simulated variables. This classifier was trained using a test ratio of 0.3. Since the goal of these models is to reproduce mutational signatures, the model's performance was tested by simulating donors and running SigProfilerExtractor[34] (v1.1.21) to detect the simulated signatures. During the training phase, we experimented with different configurations and observed that the use of fewer epochs caused SigProfilerExtractor to detect new signatures (artifacts) not present in the catalog.

### Copy number alterations and structural variants

Two independent models were trained using a CTAB-GAN+[29] architecture to simulate copy number alterations and structural variants. For CNAs, the training file includes the total length of altered segments (log scale), the copy number state of both alleles, and the tumor type to which the alteration belongs. Events with rare copy number states (< 0.5%) were excluded. The SV training file includes the genomic position where the event starts, the event length (log scale), strand information, and the SV class. For translocations, the length represents the linear distance between the two breakpoints of the chromosomes involved in the event, allowing the model to reproduce interactions between close or distant chromosomes with a frequency similar to that observed in the real data. However, the original PCAWG files did not provide information about the actual sizes of the two fragments involved in translocations.

### Augmenting DeepTumour training data

DeepTumour is a fully connected feedforward deep neural network that in its latest version classifies 29 common cancer types based on mutation positional and context distributions[37]. To generate the positional distributions, DeepTumour divides the genome into 2,897 bins (1Mbp) and counts the number of somatic mutations within each bin. Additionally, it calculates the distribution of 150 different mutation types in their single (e.g. C > G), dinucleotide (e.g. AC > AG), and trinucleotide contexts (e.g. ACA > AGA). The count for each mutation type is then normalized by the total number of SNVs in the sample and then transformed into z-scores for each feature. These positional and contextual mutation distributions result in a total of 3,047 features.

To observe changes in DeepTumour's performance with simulated samples generated by OncoGAN, we trained a new model using the same architecture as for the baseline model. Originally for the baseline model, the 2016 release of 2,778 PCAWG data was filtered for tumor types with 18 or more samples, resulting in 29 cancer types. These samples were then split into 80% training data, 10% validation data, and 10% testing data, repeated five times since the model was trained following a 5-fold cross validation. Using each of the five training partitions, the model was trained with Adam50, employing a batch size of 32 for 400 epochs in PyTorch 1.9.1 with CUDA 10.2 support. Hyperparameter optimization was conducted by improving the model's performance on the corresponding split of validation data using the 'gp_minimize' function from the scikit-optimize 0.9.0 Python library. The new model was trained using a consistent approach, incorporating 100 additional simulated samples generated by OncoGAN for each tumor type studied in the original 80% of the training data. The code was written in Python 3.7.16.

### Custom genome reconstruction

To create a custom FASTA genome with germline and somatic mutations as well as copy number alterations –suitable for generating raw sequencing data– we followed the approach described below. First, using a reference genome as a template, we duplicate each chromosome to create two alleles. Germline mutations (i.e., SNPs provided by the user) are randomly assigned to one allele (heterozygous) or both alleles (homozygous), and are incorporated into the genome before any somatic events.

Along with the VCFs for mutations, CNAs and SVs, OncoGAN provides a file specifying the order in which events occur in the tumor, including the allele in which mutations or deletions take place, and the source and target alleles for duplication events. Following this order, mutations are added to the appropriate allele, deleted regions are removed, and duplications are simulated by creating a new entry that is later merged into a tandem duplication. This approach allows mutations to be introduced independently into the original or the duplicated allele after the duplication event. Once mutations have been incorporated and alleles have been duplicated or deleted based on the CNA status, structural variants are arranged, ensuring that the correct allele is modified (Data S1). Finally, both haplotypes for each chromosome are merged into a single sequence and separated by 1,000 'N' bases to prevent artifacts.

A tracking file is generated at the beginning of the process to monitor the actual positions of mutations in the custom genome for each allele, since indels, CNAs and SVs may shift reference coordinates.

## QUANTIFICATION AND STATISTICAL ANALYSIS

### OncoGAN individual quality control metrics
#### *Donor characteristics*
To assess the similarity between real and simulated populations, we calculated the histogram intersection distance for each type of mutation using the HistogramTools (v0.3.2) R package. First, we determined the number of each mutation type per donor. Then, we generated histograms for both the real and simulated datasets using the same bins (nbin = 10, spanning from the minimum to maximum values for each mutation type). Histogram counts were normalized by the total number of donors to make the histograms comparable. We calculated three different intersection distances to evaluate the differences between the populations: i) The histogram intersection distance between the PCAWG and OncoGAN datasets. ii) A bootstrap analysis (n=1,000) for the above comparison to calculate confidence intervals. iii) A control analysis in which the PCAWG dataset was split into two random subgroups, and the intersection distance was calculated within them (n=1,000). The closer the intersection distance is to 0, the more similar the two populations are.

To compare the simulations not only to their corresponding real tumors but also to other tumor types, we calculated the histogram intersection distances across the eight studied tumor types. In this case, we used the total number of mutations per donor as the comparison metric. Using the minimum and maximum values across the eight tumor types, we defined 100 bin ranges to ensure consistency across all comparisons. We compared the PCAWG dataset against itself to determine the true relationship between the tumor types and then compared the OncoGAN dataset against PCAWG. For a more realistic comparison when calculating the intersection distance within the same tumor type for the PCAWG dataset (e.g., Breast-AdenoCa vs. Breast-AdenoCa), we randomly split the dataset into two subgroups and performed the intersection distance calculation 200 times, reporting the mean intersection distance. Finally, to evaluate the similarity between real and simulated distributions, we calculated the Pearson correlation using the mean intersection distance metric reported above between OncoGAN and PCAWG datasets for the total number of mutations.

#### *Driver genes*
To verify the accuracy of simulating driver gene co-occurrence, we calculated the percentage of donors harboring each possible 1-vs.-1 driver combination for both the real and simulated datasets. The four possible combinations were: i) neither driver was present; ii) only "driver A" was present; iii) only "driver B" was present; and iv) both drivers were present. Pearson correlation was then calculated independently for each of the eight studied tumor types, considering all combinations. The number of selected drivers and combinations can be found in Table S4.

To further assess whether the introduced mutations are identified as driver genes at the population level, as observed in real data, we used the software ActiveDriverWGS (v1.2.1)[36]. The analysis was performed independently for both the PCAWG and OncoGAN datasets, using ActiveDriverWGS's default cancer gene list and configuration. Genes were classified as drivers if their FDR-corrected p-value was below 0.05.

#### *Variant allele frequency*
To confirm that the sampling method accurately simulates the mean variant allele frequency per donor, we performed a two-sided Wilcoxon test comparing real and simulated donor VAFs independently for each tumor type. Additionally, a two-sided F-test was conducted to compare the variances of real and simulated populations. For both analyses, the number of samples in each group corresponds to the number of real (*see Data Preprocessing section above*) and simulated (n=100) donors, respectively, for each tumor type.

To validate the accuracy of the copy number-driven simulation of allele frequencies, we simulated germline mutations by extracting positions from dbSNP (v153) and assigned their VAFs using the same methodology that we applied to somatic mutations. We then manually verified that the allele frequencies of the simulated mutations matched the copy number state of each genomic region.

#### *Genomic position*
To evaluate how accurately the locations of mutations were simulated according to patterns found in real tumors, we employed three approaches: i) The human genome was divided into 3,097 bins (1Mbp) and the percentage of mutations in each bin for both the real and simulated datasets was calculated. Pearson correlation was then estimated for each of the studied tumor types using these values. ii) A linear regression was performed to compare the PCAWG dataset against itself, and OncoGAN vs. PCAWG datasets to determine whether there were differences in the simulation of mutation positions between regions with lower or higher density of mutations, reporting the $R^2$ value. When comparing the same tumor type (e.g. Breast-AdenoCa vs. Breast-AdenoCa), we split

the PCAWG dataset into two random subgroups and ran the linear regression 200 times, reporting the mean values. iii) t-SNE visualization (Rtsne package v0.17 with default parameters) was used to cluster real and synthetic donors based on the genomic distribution of their mutations (absolute values for 1Mbp windows).

### Mutational signatures

To detect mutational signatures present in OncoGAN simulations, we employed the same preprocessing approach used for the PCAWG dataset, using SigProfilerExtractor[34] (v1.1.21) with its default configuration, except the –maximum_signatures option was set to 8. Signatures were then manually compared to determine whether all expected signatures were simulated and to check for any new signatures or artifacts. We also compared the percentage of donors harboring the mutations with the corresponding percentages found in the real data. To investigate discrepancies between OncoGAN simulations and SigProfiler detections, we visualized the percentage of mutations –both simulated and detected– associated with each signature for each donor. Additionally, we examined discrepancies in the primary trinucleotide contexts ($\geq$0.5%) present in each signature by comparing COSMIC (v3.4) (https://cancer.sanger.ac.uk/signatures/downloads/) with their usage in the PCAWG training file and the dataset generated by OncoGAN. Pearson correlations were calculated to assess the similarity between the patterns using the contribution of the trinucleotide contexts for each signature.

### Indel distribution

To evaluate whether the distribution of indel sizes in the simulations closely matched that observed in the real data, we performed a Wilcoxon test between the two datasets for each indel size. For this analysis, we calculated the percentage that each indel length contributed to the total number of indels for each donor. The analysis was restricted to indels up to 5 nucleotides in length, as these account for an average of 88.5% of all indels. The sample size used for each comparison corresponds to the number of donors available for each tumor type.

### Mutational consequences

The Docker version of the Ensembl Variant Effect Predictor tool (v112)[35] was used to predict the effects of real and simulated mutations. The cache for the *Homo sapiens* GRCh37 version of the genome was downloaded from the Ensembl SFTP server (https://ftp.ensembl.org/pub/release-112/variation/indexed_vep_cache/), and results were obtained only for canonical transcripts using the following command: vep –offline –format vcf –dir_cache /cache/ –force_overwrite –total_length –numbers –ccds –canonical –biotype –pick –no_stats –assembly GRCh37. To enhance plot readability, Y axes were broken using the R package ggbreak (v0.1.2)[55].

### Copy number alterations and structural variants

The similarity between real and simulated CNA and SV profiles was compared using four chromosomal instability metrics adapted from the CINmetrics package[38]. For copy number alterations, we used the copy number abnormality score[56], which is defined as the number of segments with an altered copy number (i.e., where the major and minor alleles differ from one). Additional metrics include the mean length per altered segment, the fraction of the genome altered (calculated as the total length of altered segments normalized by genome size), and the total aberration index (TAI)[57], which is calculated as:

$$\text{TAI} = \frac{\sum(Aberrant\ segment\ length \cdot |Segment\ mean|)}{Total\ aberrant\ segment\ length}$$

Where segment mean is defined as:

$$Segment\ mean = log2\left(\frac{Total\ copy\ number}{2}\right)$$

For structural variants, two metrics were used: the number of SVs and the mean length of the SV events per donor. In all cases, the Wilcoxon test was used to compare the means of the different metrics between the real (*see Data Preprocessing section above for the number of donors*) and simulated donors (n=100).

### DeepTumour prediction

We used this version of DeepTumour as a quality control tool to predict the tumor type of origin for our simulations, providing the predicted tumor type and the probability of the sample belonging to any other possible tumor types. To explain the Eso-AdenoCa results, we performed a PCA using the PCAtools (v2.14.0) R package and the main donor characteristics (number of mutations for each type and mutational signature). For Lymph-MCLL, we compared their genomic position metrics with those from Lymph-BNHL.

To compare the performance of DeepTumour trained with simulated samples to the baseline model, we evaluated traditional performance metrics, including accuracy, precision, recall, and F1 score, independently using the corresponding test partitions. To calculate the overall accuracy, we calculated the proportion of correct classification. This metric is applied only when summarizing the performance of the classifier across all 29 tumor types. Recall was calculated as the proportion of samples from a specific cancer type that were correctly classified as that type, while precision measures the proportion of samples assigned to a particular type that truly belong to that type. The F1 score represents the harmonic mean of recall and precision. When reporting independent Lymph-MCLL and Lymph-UCLL metrics, Lymph-CLL donors and misclassified donors from other tumor types were divided based on the presence or absence of the SBS9 mutational signature, respectively.

