## [Document S2. Transparent peer review records for Diaz-Navarro et al. · Cell Genomics]

***In silico* generation of synthetic cancer genomes using generative AI**

Ander Díaz-Navarro, Xindi Zhang, Wei Jiao, Bo Wang, and Lincoln Stein

Summary

Initial submission: Received : Feb 24, 2025

Scientific editor: Sara Rohban

First round of review: Number of reviewers: 2
Revision invited : Apr 08, 2025
Revision received : Jun 06, 2025

Second round of review: Number of reviewers: 1
Accepted : Jul 14, 2025

Data freely available: YES

Code freely available: YES

This transparent peer review record is not systematically proofread, type-set, or edited. Special characters, formatting, and equations may fail to render properly. Standard procedural text within the editor's letters has been deleted for the sake of brevity, but all official correspondence specific to the manuscript has been preserved.

Referees' reports, first round of review

Reviewer #1:

The paper by Díaz-Navarro et al. tackles the important problem of generating synthetic cancer genomes. The manuscript is well-structured and gently guides the reader through the idea of factorising the complex task of genome generation into smaller subtasks, such as generating passenger mutations, driver mutations, and structural variants, and training a separate generative model for each subtask. The OncoGAN pipeline then combines the outputs of the individual models to assemble an entire cancer genome. The performance of the model is assessed by comparing various statistics of the generated genomes with the real genomes from the PCAWG dataset. The accompanying code is open-source, well-documented, and the trained models have been made available.

However, I am not entirely convinced that the particular factorisation of the genome generation task proposed in the paper fully reflects the complexity of the underlying biology.

Major comments:

1. Using flexible generative models such as GANs guarantees that data distribution for each individual subtask is modelled very accurately. However, the different models operate independently. While this will lead to realistic marginal distributions, it may lead to inconsistencies. One critical aspect is that point mutation VAFs are CNAs seem to be modelled independently of each other, even though they are very tightly connected biologically as the mutations may sit on amplified copies or in regions with LOH. This needs to be clarified and fixed.
2. In Fig. 3B, the PCAWG genomic distribution is noticeably noisier, especially for Liver-HCC. It would be interesting if the authors could explore what additional variance is not modelled by OncoGAN.
3. Expanding the comment 2, I would like to see a paragraph discussing the limitations of the method. What are the trade-offs of using such a factorised approach? What would be potential directions for future studies (besides the possibility of training on larger datasets and rare tumour types)?
4. There have been previous attempts to generate synthetic genomes. The authors should provide some quantitative comparisons with approaches like <https://www.nature.com/articles/s41587-019-0364-z> and Xome-Blender. It would be especially interesting to evaluate the performance of aspects of modelling that require the interaction of different models inside the OncoGAN pipeline (see comment 1). Alternative approaches that use real BAM files could serve as gold standards in that regard.

Minor comments:

1. While the provided description of the OncoGAN pipeline is thorough and detailed, it is, at times, difficult to follow due to unclear connections between the models. The comprehension curve would be less steep if the authors provided a "data flow" chart, similar to Fig. 1 but more detailed, as one of the Supplementary figures, along with the inference pseudocode. Additional clarifications can also be put in the text.

2. Line 492 "reported in the PCWG study" - "A" missing

Overall, this is a good study, which could be further improved by studying the limitations, comparing it to other methods and clarifying the use of the models inside the pipeline.

Reviewer #2:

Diaz-Navarro and colleagues present a novel framework for cancer genome simulation, introducing a GAN-based and VAE-based suite of data generators collectively termed OncoGAN. This framework is designed to simulate diverse classes of somatic alterations - including single nucleotide variants (both drivers and passengers), copy number alterations (CNAs), and structural variants (SVs) with the goal of generating synthetic tumour genomes that better reflect the mutational landscapes observed in real cancers.

By leveraging generative adversarial networks (GANs), OncoGAN produces simulated genomes that more closely resemble the complexity and distribution of mutations seen in actual tumour genomes, improving on existing simulators in this regard. The tool outputs variant call format (VCF) files, which can, in principle, be processed by external read simulators to generate FASTQ or BAM files. The authors demonstrate this by integrating OncoGAN with InSilicoSeq.

The OncoGAN models are trained separately on data from eight cancer types, using mutation data from the TCGA/ICGC PCAWG project. The authors validate their approach by first training independent GAN/VAE models for each mutation type and showing that the resulting synthetic genomes replicate key statistical and spatial characteristics of the real tumour data. These simulations are then evaluated for their utility in downstream tasks such as mutational signature extraction, driver detection, and chromosomal instability quantification.

Notably, the authors apply their simulated data to augment real tumour genomes in a classification task: predicting tumour type based on genome-wide mutational distributions using their previously developed framework, DeepTumour. The addition of OncoGAN-derived data leads to a modest but consistent improvement in classification accuracy, highlighting the potential of this tool for data augmentation in algorithm development.

Finally, the authors provide a dataset of 800 simulated tumour genomes, publicly available for use by the scientific community, offering a resource that circumvents some of the data-access limitations often faced in cancer genomics research.

Overall, OncoGAN represents a promising and thoughtfully designed approach to tumour genome simulation. Its emphasis on realism and public availability are particularly commendable. However, I would like to raise three major points for consideration. I encourage the authors to elaborate on these points in order to broaden the applicability and impact of this promising approach, to enhance both the transparency and the utility of the framework.

Major Comments.

Limitations of simulation realism.

OncoGAN aims to simulate a broad spectrum of mutation types in a "highly realistic" manner. The use of GANs is a strong methodological choice that clearly enhances the realism of simulated genomes.

However, accurately modeling the biological complexity underlying each mutation class is a substantial challenge. While SNVs are arguably the most tractable, even these involve nuanced biological features that are difficult to simulate comprehensively.

The authors distinguish between driver and passenger SNVs and aim to recapitulate key characteristics of passengers such as their number, trinucleotide context, and genomic distribution. This enables certain downstream applications, such as driver detection and mutational signature analysis, to benefit from the simulated data. However, other biological processes contributing to SNV landscapes appear to be unmodeled. These include dependencies on transcriptional and replicative timing, strand bias (e.g., APOBEC-related mutations), allele-specific mutation patterns, and clonal timing (e.g., early vs. late or subclonal events). The absence of these features limits the range of downstream tasks that could leverage OncoGAN simulations effectively. For instance, methods designed to detect clustered or strand-coordinated mutations.

While the utility of OncoGAN for tasks like tumour classification is clearly demonstrated, it would be helpful if the authors could more explicitly delineate which classes of downstream applications are well-supported by the current level of realism, and which may not yet be.

Modeling of CNAs and SVs.

OncoGAN includes modules for simulating CNAs and SVs, but these components appear less mature than the SNV generator, and important implementation details remain unclear.

a. Like SNVs, CNAs and SVs play crucial roles in tumourigenesis, particularly when they affect known driver genes or recur at specific genomic hotspots. It is not clear how the current simulation framework accounts for such biologically meaningful patterns, such as amplification of oncogenes, deletion of tumour suppressors, or high-frequency gain/loss peaks.

b. The timing of CNA events, particularly gains, as well as the presence of subclonal CNAs, can

significantly influence variant allele frequencies (VAFs). It is unclear to what extent the OncoGAN simulator ensures consistency between CNAs and the VAFs of the mutations they bear. The authors' example using generateBAM.py from the InSilicoSeq integration suggests that such modeling may be possible, but this is not clearly described in the manuscript.

c. Does InSilicoSeq support phasing of mutations and properly phased haplotypes in diploid genomes? This could impact the realism of simulated BAM files, especially when evaluating tools that rely on haplotype information.

Definition of "cancer genome" simulation.

While OncoGAN is described as simulating "cancer genomes", it in fact produces VCF files, which are important but partial representations of tumour genomes. The proposed way to generating full tumour BAM files involves interfacing OncoGAN with InSilicoSeq, a tool originally developed for metagenomic applications. InSilicoSeq simulates reads from a given FASTA sequence, which may not achieve the same level of realism as other read-level simulation methods that operate directly on real tumour BAM files, such as BAMSurgeon.

Although users could theoretically adapt OncoGAN outputs for use with more advanced simulators, doing so would require non-trivial additional work. It would strengthen the utility of the tool if the authors could offer guidance or future directions for improved integration with more realistic read-level simulation frameworks.

Authors' response to the first round of review

Reviewer #1

The paper by Díaz-Navarro et al. tackles the important problem of generating synthetic cancer genomes. The manuscript is well-structured and gently guides the reader through the idea of factorising the complex task of genome generation into smaller subtasks, such as generating passenger mutations, driver mutations, and structural variants, and training a separate generative model for each subtask. The OncoGAN pipeline then combines the outputs of the individual models to assemble an entire cancer genome. The performance of the model is assessed by comparing various statistics of the generated genomes with the real genomes from the PCAWG dataset. The accompanying code is open-source, well-documented, and the trained models have been made available.

However, I am not entirely convinced that the particular factorisation of the genome generation task proposed in the paper fully reflects the complexity of the underlying biology.

Major comments

1.- Using flexible generative models such as GANs guarantees that data distribution for each individual subtask is modelled very accurately. However, the different models operate independently. While this will lead to realistic marginal distributions, it may lead to inconsistencies. One critical aspect is that point mutation VAFs and CNAs seem to be modelled independently of each other, even though they are very tightly connected biologically as the mutations may sit on amplified copies or in regions with LOH. This needs to be clarified and fixed.

We thank the reviewer for pointing out this issue. We have implemented a new approach to simulate CNA-driven variant allele frequencies. This new approach and its results are described in the “Variant allele frequency” sections of the “Results” and “Methods” (pages 7, 22, and 25). Briefly, allele frequencies are simulated by modeling the sequential order of somatic mutations and copy number events. Mutations and CNAs are shuffled to mimic a random timeline of events, and variant allele frequencies are then computed based on the resulting allelic composition. To validate this approach, we also simulated germline mutations using SNPs from dbSNP as input and computed their VAFs using the same method (in this case, germline mutations are introduced before any CNA event). The results are shown in Figure 1 (Supplementary Figure S6), where germline mutation allele frequencies reflect the underlying CNAs. As part of this new implementation, OncoGAN also returns the file specifying the order of events used to simulate allele frequencies. This file is later used to accurately simulate synthetic BAMs (Supplementary Figure S22), as further detailed in our response to Reviewer #2.

In addition, we have decided to retain the original method for simulating VAFs in simpler cases where the user does not need or wish to simulate CNA-SV events—for example, to generate a range of clonal and subclonal frequencies suitable for benchmarking variant calling or for mutational signature analysis.

Figure 1. Quality control plots for validating the accuracy of CNA-driven allele frequency simulation. A) Copy number alteration profiles for three chromosomes, each showing different combinations of copy number events. B) Variant allele frequency distributions of germline and somatic mutations, reflecting the underlying CNA pattern. Points are colored by the number of alleles in which each mutation appears.

2.- In Fig. 3B, the PCAWG genomic distribution is noticeably noisier, especially for Liver-HCC. It would be interesting if the authors could explore what additional variance is not modelled by

OncoGAN.

As pointed out by the reviewer, the OncoGAN-generated Liver-HCC data shows a less noisy distribution in Figure 3B of the manuscript, which relates to the genomic distribution of mutations across the genome. First, we explored whether other features exhibited the same behavior. In Supplementary Figure S2, we show the number of each mutation type in real and simulated donors across the eight tumor types. The simulated mutation density closely resembles that of the real donors. However, in the

case of Liver-HCC, while the simulated densities approximate the real ones, extreme values in the total number of mutations (very low or very high) are not simulated with the same frequency. This explains why the intersection distance in Supplementary Figure S3 for Liver-HCC is higher when comparing simulated vs. real donors than when comparing two populations within the PCAWG dataset.

As explained in the “Mutation density” section (page 7), this discrepancy may be attributable to the large sample size for Liver-HCC ($n = 264$), which can result in an imbalanced dataset where donors with unusually low or high numbers of mutations are underrepresented. For the other tumor types, the metrics between real and synthetic donors are more similar. Notably, the second tumor type with the highest variance between real and synthetic donors is Prost-AdenoCa (Supplementary Figure S3), which also has the second-largest number of donors. This suggests that donor imbalance may be responsible for the discrepancies observed.

To confirm this, we reviewed the mutation pattern in Supplementary Figure S8 for Liver-HCC. Although it is generally well captured (Pearson correlation = 0.887), some specific peaks differ between real and synthetic distributions. We explored this further by analyzing the variance between real and simulated donors Figure 2 (Supplementary Figures S9-S10). For Liver-HCC, we observe specific regions where the variance is not well simulated (Figure 2A - Supplementary Figure S9), which is corroborated by the Manhattan plot in Figure 2B (Supplementary Figure S10). Although Eso-AdenoCa also shows variance differences (Supplementary Figures S9-S10), these are randomly distributed across the genome and do not affect consecutive windows, whereas Liver-HCC presents consistent, localized discrepancies spanning larger and specific regions. The lack of variance simulation in these highly variable regions may explain why the synthetic LiverHCC dataset appears less noisy in Figure 3B of the manuscript.

Figure 2. Analysis of mutation density variance across genomic regions in real and synthetic Liver-HCC donors. A) Variance ratio between PCAWG and OncoGAN donors, indicating a few specific regions where variance is not well simulated. B) Manhattan plot highlighting regions with significant variance

differences. Red dots represent regions with FDR values below 0.005. We attempted to find an explanation by examining the composition of Liver-HCC donors, as three different ICGC studies contributed to this tumor type in PCAWG: LICA-FR, LINC-JP, and LIRIJP. However, the vast majority of donors come from LIRI-JP ($n = 230$), and when we compared genomic variance across the three datasets, no distinct regions were identified. Therefore, we believe the reduced noise results from the same imbalance issue, where only a few donors show high variability and the model captures more of the average variance across the entire dataset.

These new findings have been incorporated into the manuscript under the “Genomic distribution” section on page 9.

3.- Expanding the comment 2, I would like to see a paragraph discussing the limitations of the method. What are the trade-offs of using such a factorised approach? What would be potential directions for future studies (besides the possibility of training on larger datasets and rare tumour types)?

We have expanded the “Discussion” section to address the limitations and trade-offs of our factorized approach (page 19). In summary, OncoGAN currently cannot simulate subclonal architectures or the interplay between driver mutations and copy number alterations, as these elements are modeled independently. Additional limitations arise from the biological complexity of tumors and the lack of paired datasets needed to capture features such as chromatin state, replication timing, or strand bias – as pointed out by Reviewer #2– which are only partially inferred and not explicitly simulated.

4.- There have been previous attempts to generate synthetic genomes. The authors should provide some quantitative comparisons with approaches like <https://www.nature.com/articles/s41587-0190364-z> and Xome-Blender. It would be especially interesting to evaluate the performance of aspects of modelling that require the interaction of different models inside the OncoGAN pipeline (see comment 1). Alternative approaches that use real BAM files could serve as gold standards in that regard.

We attempted to follow the reviewer’s suggestion and compare OncoGAN with the two referenced approaches. However, both tools were specifically designed to generate synthetic data for benchmarking subclonal reconstruction methods. As explained in the discussion, a key limitation of OncoGAN is that it does not currently simulate subclonal architecture, making a direct comparison with these methods unfeasible. We acknowledge this limitation, and in the “Discussion” section we suggest that users interested in subclonal reconstruction refer to these two approaches (page 19).

Additionally, the first method, described by Salcedo et al., 2020, uses BAMsurgeon to spike mutations into real sequencing data from the Genome-in-a-Bottle consortium. These mutations follow replication patterns derived from exome sequencing data and known mutational signatures, producing realistic but tumor-agnostic profiles. Xome-Blender (Semeraro et al., 2018), on the other hand, generates somatic

mutations randomly. In both cases, the focus is not on reproducing tumor-type-specific genomic characteristics, which is the central goal of OncoGAN. Therefore, a quantitative comparison would not be meaningful due to the differing objectives of the tools.

Nevertheless, as both reviewers suggested, leveraging BAM-level simulators may produce more realistic outputs than directly simulating FASTQ files. To this end, we integrated BAMsurgeon into our pipeline to apply OncoGAN's mutation profiles to real input BAMs. While we successfully incorporated the original BAMsurgeon tool

(<https://github.com/adamewing/bamsurgeon>), we were unable to install its subclonal simulation module due to known installation issues

(<https://rt.cpan.org/Public/Dist/Display.html?Name=NGS-Tools-BAMSurgeon>). Xome-Blender was not adopted, as it does not support custom mutation input, making it incompatible with OncoGAN's outputs. The implementation of BAMsurgeon in our workflow is described in the "Compatibility with FASTQ and BAM generation tools" section (page 17).

Minor comments

1.- While the provided description of the OncoGAN pipeline is thorough and detailed, it is, at times, difficult to follow due to unclear connections between the models. The comprehension curve would be less steep if the authors provided a "data flow" chart, similar to Fig. 1 but more detailed, as one of the Supplementary figures, along with the inference pseudocode. Additional clarifications can also be put in the text.

We appreciate the reviewer's suggestion and have implemented some clarifications in the "MultiModel Ensemble Pipeline for Synthetic Tumor Generation" section (pages 4-6) to make the pipeline easier to follow and to clearly indicate which models are used at each step. In addition, we have added a new supplementary figure (Supplementary Figure S1), which presents a pseudocode overview of the pipeline along with example outputs from each model, illustrating how they are connected. This simplified version is intended to convey the main logic of the inference process; however, if the reviewer feels that any specific part requires further detail, we would be happy to elaborate.

2.- Line 492 "reported in the PCWG study" - "A" missing

We have fixed the typo.

Overall, this is a good study, which could be further improved by studying the limitations, comparing it to other methods and clarifying the use of the models inside the pipeline.

Reviewer #2

Diaz-Navarro and colleagues present a novel framework for cancer genome simulation, introducing a GAN-based and VAE-based suite of data generators collectively termed OncoGAN. This framework is designed to simulate diverse classes of somatic alterations - including single nucleotide variants (both drivers and passengers), copy number alterations (CNAs), and structural variants (SVs) with the goal of generating synthetic tumour genomes that better reflect the mutational landscapes observed in real cancers.

By leveraging generative adversarial networks (GANs), OncoGAN produces simulated genomes that more closely resemble the complexity and distribution of mutations seen in actual tumour genomes, improving on existing simulators in this regard. The tool outputs variant call format (VCF) files, which can, in principle, be processed by external read simulators to generate FASTQ or BAM files. The authors demonstrate this by integrating OncoGAN with InSilicoSeq.

The OncoGAN models are trained separately on data from eight cancer types, using mutation data from the TCGA/ICGC PCAWG project. The authors validate their approach by first training independent GAN/VAE models for each mutation type and showing that the resulting synthetic genomes replicate key statistical and spatial characteristics of the real tumour data. These simulations are then evaluated for their utility in downstream tasks such as mutational signature extraction, driver detection, and chromosomal instability quantification.

Notably, the authors apply their simulated data to augment real tumour genomes in a classification task: predicting tumour type based on genome-wide mutational distributions using their previously developed framework, DeepTumour. The addition of OncoGAN-derived data leads to a modest but consistent improvement in classification accuracy, highlighting the potential of this tool for data augmentation in algorithm development.

Finally, the authors provide a dataset of 800 simulated tumour genomes, publicly available for use by the scientific community, offering a resource that circumvents some of the data-access limitations often faced in cancer genomics research.

Overall, OncoGAN represents a promising and thoughtfully designed approach to tumour genome simulation. Its emphasis on realism and public availability are particularly commendable. However, I would like to raise three major points for consideration. I encourage the authors to elaborate on these points in order to broaden the applicability and impact of this promising approach, to enhance both the transparency and the utility of the framework.

Major comments

1.- Limitations of simulation realism.

OncoGAN aims to simulate a broad spectrum of mutation types in a "highly realistic" manner. The use of GANs is a strong methodological choice that clearly enhances the realism of simulated genomes. However, accurately modeling the biological complexity underlying each mutation class is a substantial challenge. While SNVs are arguably the most tractable, even these involve nuanced biological features that are difficult to simulate comprehensively.

The authors distinguish between driver and passenger SNVs and aim to recapitulate key characteristics of passengers such as their number, trinucleotide context, and genomic distribution. This enables certain downstream applications, such as driver detection and mutational signature analysis, to benefit from the simulated data. However, other biological processes contributing to SNV landscapes appear to be unmodeled. These include dependencies on transcriptional and replicative timing, strand bias (e.g., APOBEC-related mutations), allele-specific mutation patterns, and clonal timing (e.g., early vs. late or subclonal events). The absence of these features limits the range of downstream tasks that could leverage OncoGAN simulations effectively. For instance, methods designed to detect clustered or strand-coordinated mutations.

While the utility of OncoGAN for tasks like tumour classification is clearly demonstrated, it would be helpful if the authors could more explicitly delineate which classes of downstream applications are well-supported by the current level of realism, and which may not yet be.

We would like to thank the reviewer for pointing out some limitations we had not fully considered when we submitted the manuscript. As also suggested by Reviewer #1, we have expanded the "Discussion" section (page 19) to include a more detailed overview of the current limitations of OncoGAN and a list of suitable downstream applications for the current version of OncoGAN.

2.- Modeling of CNAs and SVs.

OncoGAN includes modules for simulating CNAs and SVs, but these components appear less mature than the SNV generator, and important implementation details remain unclear.

- A. Like SNVs, CNAs and SVs play crucial roles in tumorigenesis, particularly when they affect known driver genes or recur at specific genomic hotspots. It is not clear how the current simulation framework accounts for such biologically meaningful patterns, such as amplification of oncogenes, deletion of tumour suppressors, or high-frequency gain/loss peaks.

We agree that CNAs and SVs play a critical role in tumorigenesis and that faithfully modeling these alterations is essential for realistic tumor genome simulation. To address this, we have incorporated a method that simulates high-frequency CNAs—referred to as driver CNAs—which is explained in the Results and Methods sections (pages 7 and 23, respectively). Using a simple approach, we aggregated

the copy number states of cytobands from the PCAWG dataset for each tumor type and, during simulation, sample events based on their observed frequency in the original dataset.

We acknowledge that this approach has limitations, as it does not account for the correlation patterns between different CNA driver events (unlike what we do for driver mutations), due to insufficient data to train such a model. Moreover, while high-frequency CNAs may represent driver events, there are no curated lists of CNA drivers available from the PCAWG dataset. As a result, we used a recurrence threshold ($\geq 5\%$) to define driver CNAs, assuming that such events are relevant and should be replicated at similar frequencies in the simulated donors.

To ensure variability in size, sampled driver CNAs are combined with randomly simulated events, allowing boundaries to be adjusted based on surrounding alterations, rather than having all driver CNAs with identical lengths.

Unfortunately, we were unable to apply a similar strategy to SVs, as we did not have sufficient data to obtain reliable patterns.

- B. The timing of CNA events, particularly gains, as well as the presence of subclonal CNAs, can significantly influence variant allele frequencies (VAFs). It is unclear to what extent the OncoGAN simulator ensures consistency between CNAs and the VAFs of the mutations they bear. The authors' example using `generateBAM.py` from the InSilicoSeq integration suggests that such modeling may be possible, but this is not clearly described in the manuscript.

As we explained in our response to Reviewer #1 – Comment 1, we have implemented a new approach to model the variant allele frequency of mutations based on copy number alteration events. To achieve this, OncoGAN now generates a file detailing the order in which mutations and CNAs occur in the tumor genome, including information about the allele on which each event takes place and, in the case of duplications, the source and destination alleles.

To improve compatibility with other tools, we have divided this process into two steps. First, the “OncoGAN-to-FASTA” function uses OncoGAN’s output along with a custom set of germline SNPs (optionally provided by the user) to create a tumor FASTA genome containing all the alterations. Mutations are added following the previously simulated event timing, ensuring that the resulting VAFs are consistent with the underlying CNA profile.

Second, the “InSilicoSeq” function –a wrapper around InSilicoSeq– uses this custom FASTA as input to generate FASTQ reads. Although we recommend InSilicoSeq for this step, the custom FASTA can be used with any other FASTA-to-FASTQ read simulator.

This functionality is described in more detail in the “Compatibility with FASTQ and BAM generation tools” section of the Results (page 16) and in the newly added Methods section titled “Custom genome reconstruction” (page 28). Furthermore, instructions for using this functionality are also provided on our GitHub.

C. Does InSilicoSeq support phasing of mutations and properly phased haplotypes in diploid genomes? This could impact the realism of simulated BAM files, especially when evaluating tools that rely on haplotype information.

InSilicoSeq, like any other FASTQ generator that uses a FASTA genome as input, does not support phasing mutations directly, as it simply generates reads from the given sequence. However, it can preserve phasing if the input FASTA already contains phased mutations. As explained above, the “OncoGAN-to-FASTA” function addresses this by starting from a diploid genome and assigning mutations to the corresponding simulated allele.

To evaluate whether our proposed approach preserves phasing correctly, we conducted a test in a 20 Mb region of the genome. In this region, we introduced a copy number alteration event and several closely located mutations to visually assess haplotypes. As shown in Figure 3 (Supplementary Figure S22), after using InSilicoSeq on both FASTA reconstruction options –i.e., with and without CNA-SV events– we observed the distinct haplotypes originally simulated, along with the expected VAFs.

Figure 3. Alignment screenshot of the in silico BAMs generated by integrating OncoGAN with InSilicoSeq using two approaches: a simple one, where OncoGAN simulates only a VCF with mutations,

and a more complex one that includes both mutations and CNAs. Phased mutations are visible, and the variant allele frequency is influenced by the copy number status and the order of events.

3.- Definition of "cancer genome" simulation.

While OncoGAN is described as simulating "cancer genomes", it in fact produces VCF files, which are important but partial representations of tumour genomes. The proposed way to generating full tumour BAM files involves interfacing OncoGAN with InSilicoSeq, a tool originally developed for metagenomic applications. InSilicoSeq simulates reads from a given FASTA sequence, which may not achieve the same level of realism as other read-level simulation methods that operate directly on real tumour BAM files, such as BAMSurgeon.

Although users could theoretically adapt OncoGAN outputs for use with more advanced simulators, doing so would require non-trivial additional work. It would strengthen the utility of the tool if the authors could offer guidance or future directions for improved integration with more realistic read-level simulation frameworks.

We appreciate this suggestion to improve the usability of OncoGAN and make it more accessible to a wider range of users. In addition to the two functions explained above –OncoGAN-to-FASTA, which integrates mutations and CNA-SV events into a custom FASTA genome compatible with any FASTA-to-FASTQ generator, and its integration with InSilicoSeq– we have also implemented support for BAMSurgeon. Specifically, we developed a wrapper function that adapts OncoGAN's output for use with BAMSurgeon (page 16). While BAMSurgeon is designed to work with real BAM files, our GitHub repository includes instructions on how users can employ our InSilicoSeq wrapper to generate a basic BAM file in cases where no real BAM is available.

One limitation of this integration is that BAMSurgeon does not allow users to predefine the alleles on which mutations appear, as haplotype assignment is handled internally. In such cases, users rely on BAMSurgeon's performance, which has been extensively validated.

With the addition of the tumor FASTA reconstruction and BAMSurgeon integration, we hope to make OncoGAN more user-friendly for a variety of downstream tasks. However, we acknowledge that other simulators may require adjustments to fully incorporate OncoGAN's outputs. While our output formats are standard and easy to manipulate, we invite users to open an issue on our GitHub repository should they need assistance with integration.

Finally, we would like to clarify our choice of InSilicoSeq, which, as the reviewer noted, was originally designed for metagenomic applications. We opted for this tool over others, such as Sandy, because InSilicoSeq provides a range of error and fragmentation size models tailored to different sequencers, which are independent of the organism being simulated. Importantly, InSilicoSeq allows users to assign different abundances to each entry in a multi-FASTA file. While this is useful in metagenomics for simulating organisms at different frequencies, we leverage this feature to simulate multiple copies of

the same chromosome at varying frequencies. This enables the simulation of subclonal allele frequencies, particularly in scenarios where CNAs are not available to guide the process.

Referees' reports, second round of review

Reviewer #2:

I would like to thank the authors for their thoughtful and constructive responses to my comments, and to congratulate them on this excellent piece of work. The addition of support for CNA-SV, CNA drivers/high frequency, the enhancements to BAM simulations - including compatibility with BAMSurgeon - and the valuable discussion of the method's limitations make this a significant contribution to the literature on realistic tumour genome simulation. Importantly, the well-documented github repository and the provided containers will help users leverage and participate in improving oncoGAN in the future.